# CASPI: collaborative photon processing for active single-photon imaging

Jongho Lee[1] ✉, Atul Ingle [2], Jenu V. Chacko[3,4], Kevin W. Eliceiri [3,4,5,6,7,8] & Mohit Gupta[1,8]

Image sensors capable of capturing individual photons have made tremendous progress in recent years. However, this technology faces a major limitation. Because they capture scene information at the individual photon level, the raw data is sparse and noisy. Here we propose CASPI: Collaborative Photon Processing for Active Single-Photon Imaging, a technology-agnostic, application-agnostic, and training-free photon processing pipeline for emerging high-resolution single-photon cameras. By collaboratively exploiting both local and non-local correlations in the spatio-temporal photon data cubes, CASPI estimates scene properties reliably even under very challenging lighting conditions. We demonstrate the versatility of CASPI with two applications: LiDAR imaging over a wide range of photon flux levels, from a sub-photon to high ambient regimes, and live-cell autofluorescence FLIM in low photon count regimes. We envision CASPI as a basic building block of general-purpose photon processing units that will be implemented on-chip in future single-photon cameras.

Active imaging, where a camera is operated in unison with a controllable illumination source (e.g., a pulsed laser), enables the estimation of various scene properties in a wide range of applications (Fig. 1a). Some examples include light detection and ranging (LiDAR)[1–5], spectral fluorescence microscopy[6], fluorescence lifetime imaging microscopy (FLIM)[7–12], transient imaging[13–15], imaging through scattering[16,17], and biomedical applications[18]. In order to estimate scene features, active imaging systems frequently require precise measurements of the time-varying light intensities at each location in the scene. For example, FLIM detects tissue pathology (e.g. malignant vs. healthy tissue) by monitoring the fine-grained (~nanosecond-scale) temporal decay of fluorescence emission. Single-photon LiDAR estimates 3D scene structures in robotics, computer vision and autonomous driving applications with millimeter-to-centimeter depth resolution. These applications require photon timing information to be captured with sub-nanosecond precision.

Single-photon cameras (SPCs) are rapidly becoming the technology of choice in active imaging due to their high sensitivity to individual photons and their ability to time-tag photon arrivals with nano-to-picosecond resolution[19]. Unlike conventional cameras, SPCs enable image sensing at the fundamental limit imposed by the physics of light: an individual photon. The time-varying photon flux incident on each pixel of an SPC is measured by a histogram of photon counts as a function of detection time. We call this histogram as a 1D photon transient. Examples of the ground-truth photon flux and measured photon transients for single-photon LiDAR are shown in Fig. 1b. By raster-scanning or flood-illuminating the scene with a pulsed laser source, we obtain a 3D photon transient cube, where various scene property estimates such as depth maps and fluorescence lifetime images can be obtained as shown in Fig. 1c, d.

Despite their high time resolution, SPCs can operate reliably over only a narrow range of incident flux levels as shown in Fig. 1b. If there

[1]Department of Computer Sciences, University of Wisconsin-Madison, Madison, WI, USA. [2]Department of Computer Science, Portland State University, Portland, OR, USA. [3]Laboratory for Optical and Computational Instrumentation, University of Wisconsin-Madison, Madison, WI, USA. [4]Center for Quantitative Cell Imaging, University of Wisconsin-Madison, Madison, WI, USA. [5]Morgridge Institute for Research, Madison, WI, USA. [6]Department of Biomedical Engineering, University of Wisconsin-Madison, Madison, WI, USA. [7]Department of Medical Physics, University of Wisconsin-Madison, Madison, WI, USA. [8]McPherson Eye Research Institute, Madison, WI, USA. ✉e-mail: jlee567@wisc.edu

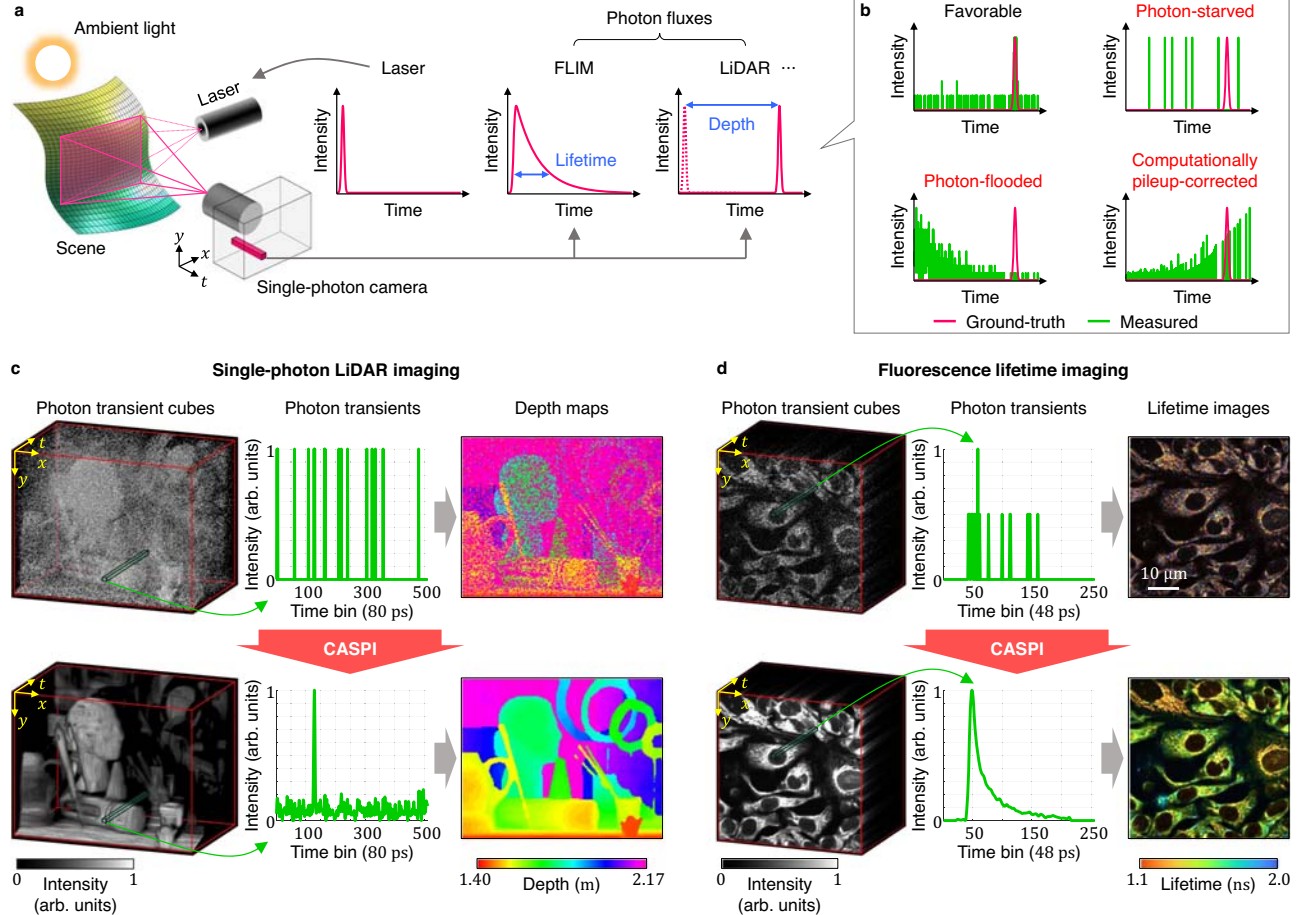

**Fig. 1 | CASPI is a versatile photon data processing technique for active imaging applications. a** In active imaging, a camera operates in synchronization with a controllable light source (e.g., a laser) to probe various scene properties such as fluorescence lifetimes and 3D depths. **b** A single-photon camera-based active imaging system can operate reliably over a limited range of photon flux levels. In low signal conditions, it suffers from strong noise due to poor signal-to-noise ratio, whereas in high illumination conditions, it suffers from severely distorted measurements, resulting in large errors in estimated depths and fluorescence lifetimes. **c**, **d** We propose CASPI, a versatile photon processing method that enables reliable scene property estimation in such challenging lighting conditions for a wide range of applications including (**c**) 3D scene recovery and (**d**) fluorescence lifetime microscopy.

are very few photons (photon-starved regime), SPC pixels suffer from unreliable estimates due to poor signal-to-noise ratios (SNRs)[20,21]. If there are too many photons (photon-flooded regime), the measured photon transients suffer from severe non-linear distortions called pileup[22–28]. Although pileup can be mitigated computationally[29], it may worsen the noise as shown in Fig. 1b. Many mission-critical active imaging applications in the real world encounter a very wide dynamic range of photon flux levels. For example, a single-photon flash LiDAR system that illuminates the entire field-of-view needs to recover 3D geometry with very few signal photons, often overwhelmed by bright sunlight. Although SPCs based on single-photon avalanche diode (SPAD) technology are rapidly becoming commercially available[30–33], the lack of a robust and versatile photon data processing pipeline may severely limit the wider deployment of this exciting technology.

Conventional image and video processing algorithms are not designed for binary photon data, and thus fail to recover photon transient cubes under challenging illumination conditions. This is because sparse binary photon counts under photon-starved regimes make it challenging to find spatio-temporal correlations (both local and non-local), which several conventional image processing techniques rely on. Applying conventional filtering algorithms after scene property estimation often fails due to severe noise that does not follow conventional noise models (see Supplementary Results). Modern deep-learning-based approaches often do not generalize well for out-of-distribution settings, making practical deployment for mission-critical applications such as biomedical imaging challenging[34]. Recent quanta image processing approaches[35–37] have shown promising results for recovering high-quality 2D intensity images for passive imaging in low signal and high dynamic range conditions. However, these methods are not applicable to robust recovery of 3D photon transient cubes in active imaging applications where the raw photon data is captured at much finer (nano-to-picosecond) time scales. Although numerous state-of-the-art approaches for active imaging[23,24,38–44] have shown varying degrees of success in specific applications, over a narrow set of operating conditions, a unifying method towards realizing a general-purpose photon processing unit (PPU) for SPCs, akin to image processing units (IPUs) in conventional CMOS cameras does not exist to date.

Here we demonstrate a photon processing technique that enables reliable scene property estimation over a wide range of operating conditions while requiring no training and remaining agnostic to the applications. We call this technique CASPI (Collaborative photon processing for Active Single-Photon Imaging). We show the benefits of CASPI through extensive simulations and real experiments for two popular applications of SPCs: single-photon LiDAR[38] and FLIM[45]. We show robust depth estimation in sub-photon regimes (<1 signal photons per pixel) and under strong background illumination (>200× higher ambient photons than signal photons). We also demonstrate 5×

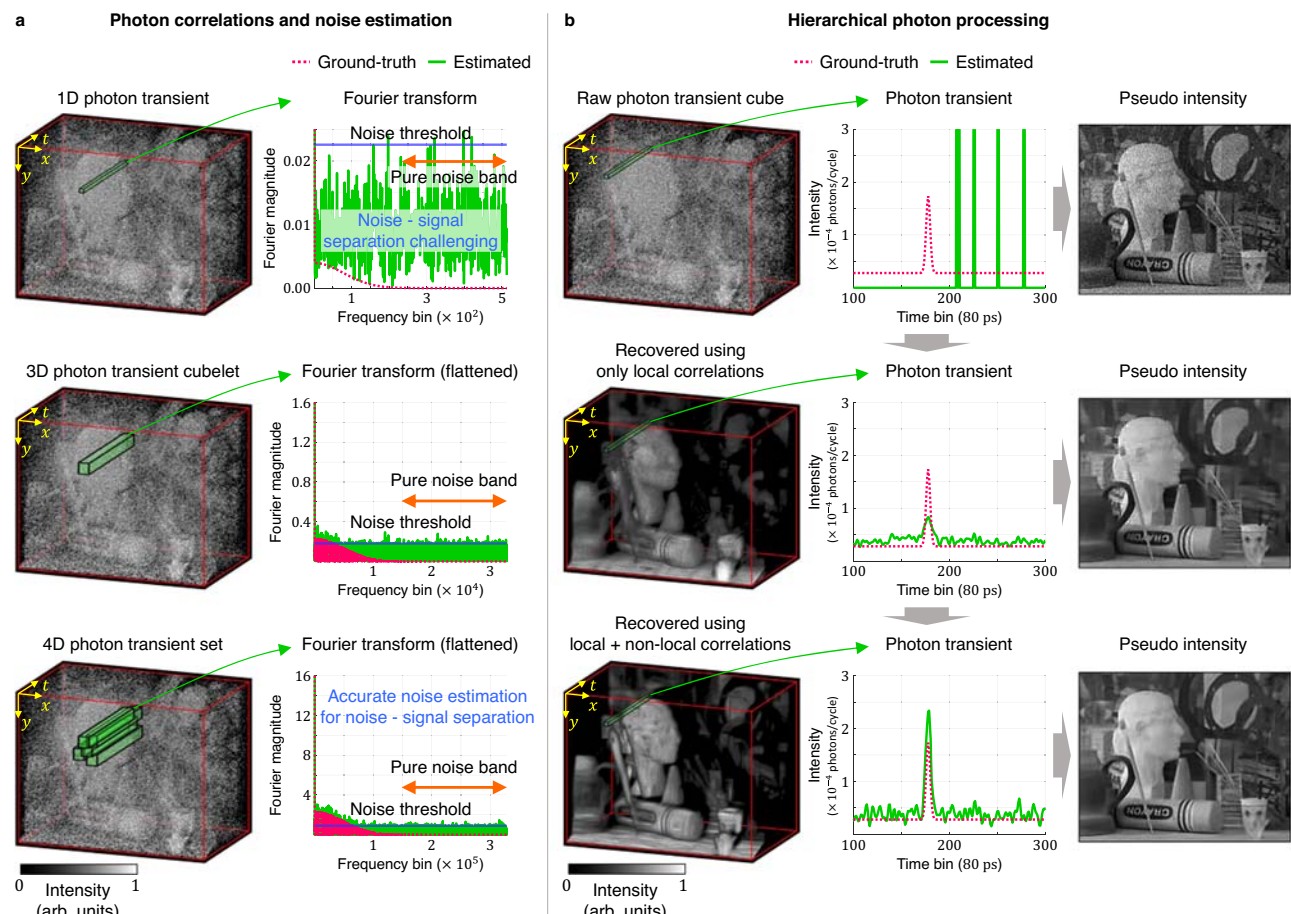

**Fig. 2 | Spatio-temporal photon correlations and hierarchical blind photon processing. a** When stronger correlations are available in the photon data, we can suppress noise effectively relative to the signal in the Fourier domain, leading to better signal and noise separation by accurate noise estimation. **b** It is challenging to use non-local correlations directly in challenging illumination conditions due to severe noise. To address this problem, we propose a hierarchical approach: We recover photon fluxes using only local correlations first, and after finding similar cubelets, final photon fluxes are recovered by exploiting local and non-local correlations collaboratively.

improvement in fluorescence lifetime estimation accuracy over state-of-the-art with as few as 10 photons per pixel (generally require 100 photons per pixel for mono-exponential decays[46]), enabling live-cell autofluorescence imaging in photon-starved regimes.

## Results

### Spatio-temporal photon correlations

CASPI relies on the following observations: (1) Photon transient cubes for most natural scenes captured by high-resolution SPCs contain abundant spatio-temporal correlations over multiple scales and dimensions, and (2) by exploiting local and non-local photon correlations collaboratively, we can recover the true photon fluxes even under challenging lighting conditions. Figure 2a shows the magnitudes of the Fourier transforms, flattened along one dimension for visualization, of different hierarchical subsets of the photon transient cube—1D photon transients, 3D photon transient cubelets and 4D photon transient sets, going from the finest to the coarsest level in the hierarchy. When correlations increase with the dimension of the subsets, the amount of noise relative to the signal decreases. This is because any structured low-frequency signal components accumulate better than random noise components in a higher dimensional Fourier transform. The noise components can be quite strong if only the 1D photon transients are considered at the finest level in this hierarchy. It is significantly reduced when not only local correlations but also non-local correlations are available in the 4D photon transient sets.

### Hierarchical blind photon processing

We take a hierarchical approach to find the similar cubelets even in the presence of strong noise and distortions as shown in Fig. 2b. First, we recover the flux estimates using only local correlations within the cubelet. Although non-local correlations are not exploited, SNRs are sufficiently improved to locate similar cubelets over larger non-local neighborhoods. After finding the similar cubelets, true photon fluxes are recovered using local and non-local correlations collaboratively from the set of similar cubelets. See "Methods" for details.

We recover the latent photon fluxes in this hierarchical photon processing by generalizing the state-of-the-art filtering framework (BM3D[47], BM4D[48], and V-BM4D[49]) to photon transient cubes. Direct generalization of these strategies, however, often fails in active imaging applications since they produce optimal results only if reliable noise statistics are available (see Supplementary Discussion). Hand-tuning of noise parameters is not feasible for many active imaging scenarios where the local SNRs change dynamically due to spatially and temporally varying illumination conditions. CASPI automatically adapts to these challenging scenarios by performing blind photon processing with accurate and independent noise estimation without requiring any prior knowledge of noise statistics.

Our key idea for accurate noise estimation is to isolate the pure noise band on the temporal frequency domain, where the noise amount can be estimated precisely as shown in Fig. 2a. The pure noise band is defined based on the following observations: (1) The noise-free incident photon fluxes at the sensor cannot contain higher frequencies than the

laser pulse since the optical path from the laser source to the sensor acts as a low-pass filter, and (2) the signal of interest (e.g., the laser pulse) spans a subset of the low frequencies since most hardware components of the laser source have limited bandwidth. We define the pure noise band as the range of frequencies where the Fourier magnitude of the laser pulse is negligibly small. After the pure noise band is isolated, the noise threshold is defined as a statistical upper bound of the magnitude in the pure noise band (see "Methods" for details).

After obtaining accurate local noise estimates, we recover the photon fluxes in two stages: initial flux estimation and final flux estimation with Wiener filtering. Wiener filtering is known to be optimal in a mean-squared-error sense if reliable SNR is available[50]. We estimate this SNR based on initial flux and noise estimates (see "Methods" for details). Noise estimation plays an important role in these flux recovery stages in the following ways: (1) In initial flux estimation using only local correlations, CASPI can recover high-frequency signal components with accurate noise thresholding in the frequency domain if the amount of noise relative to the signal is sufficiently reduced by local correlations (Fig. 2). This is different from naïve spatial binning which only uses local spatial averaging limited to the low-frequency (DC) components. (2) In initial flux estimation using both local and non-local correlations, if we retain only the low-frequency components of the set of similar cubelets, the recovered fluxes will be blurred due to the structural differences between the similar cubelets. On the other hand, if too many high-frequency components are allowed, noise will not be suppressed effectively. CASPI performs precise and effective separation of the signal and noise components in the frequency domain through accurate noise estimation to preserve high-frequency information while suppressing the noise effectively. (3) Since Wiener filtering shrinks the Fourier coefficients according to the SNRs, its performance depends on the noise and initial flux estimates.

### Guided photon processing

In very low SNR regimes, the noise components will still dominate in the transform domain notwithstanding high dimensional photon data. Therefore, it is challenging to separate the noise and signal components even with accurate noise estimation, and initial flux estimation will fail with simple thresholding in the frequency domain. We propose guided photon processing which uses spatial frequency correlations between the intensity and the photon transient cube to recover the fluxes under such low SNR scenarios. Our key insight is that a 2D pseudo-intensity image obtained by summing over the temporal dimension of the 3D photon transient cube shares the similar spatial distribution of Fourier magnitude as the 3D photon transient cube, but has significantly higher SNR due to temporal averaging.

Guided photon processing and thresholding have their own pros and cons in initial flux estimation. Although guided photon processing reduces noise more effectively in low SNR regimes, it also removes signal details. Thresholding preserves signal details better than guided photon processing in relatively high SNR regimes. CASPI selects between thresholding and guided photon processing adaptively depending on the SNR to estimate initial fluxes. See "Methods" for more details.

### Scene intensity images with CASPI

As a by-product of recovering the latent photon fluxes, CASPI can also be used for reconstructing high-quality 2D reflectance (intensity) images by integrating over the temporal dimension of the recovered 3D photon transient cube. Furthermore, if a high-quality intensity image is available as additional input from another co-located image sensor (e.g., a low-light camera), it can be used instead of the pseudo-intensity information when searching for similar photon cubelets, and in the guided photon processing step of our algorithm. All results in

this paper were generated without using intensity as additional input except for Fig. S3 in Supplementary Results where we show examples of improved depth estimation when an intensity image is available as a side input.

### High dynamic range single-photon LiDAR

We demonstrate the strengths of CASPI in a single-photon LiDAR application under a wide range of challenging illumination conditions. We quantify the illumination conditions using two different signal-to-background (SBR) metrics. $SBR_{total} = N_{sig}/N_{bkg}$ is the ratio between the average incident signal and background photons over the entire laser cycle duration, summed over all the laser cycles. $SBR_{total}$ is reported as a numeric ratio[41,51]. $SBR_{pulse} = N_{sig}/n_{bkg}$ ratio considers only those background photons that arrive during the laser pulse duration[52,53] ($n_{bkg} = N_{bkg} \times$ pulse width/laser cycle period). For a Gaussian pulse shape, the laser pulse duration is measured in terms of its full-width at half-maximum (FWHM). For $SBR_{total}$, it is important to consider background photons over the entire laser cycle since background photons that arrive earlier than the laser pulse (e.g., in high background regimes) increase the likelihood of false signal peak detection causing pileup distortion[22,23,25,29,43]. Pileup is caused by the unique sensing model of single-photon camera pixels, which, due to a finite dead-time, often only capture the first incoming photon and reject subsequent photons in each cycle. We represent $SBR_{pulse}$ as a single number but $SBR_{total}$ with two explicit photon counts, $N_{sig}$ and $N_{bkg}$. For example, even though $N_{sig}/N_{bkg} = 1/1$ and $N_{sig}/N_{bkg} = 1000/1000$ are the same $SBR_{total}$ value of 1.0, these are quite different illumination conditions due to large difference in the number of background photons. The latter can cause strong pileup distortions, resulting in large and systematic depth errors. We use the incident photon definition of SBR because incident photon counts are a more reliable indicator of the true scene illumination conditions as compared to the detected photon counts that saturate in high photon flux settings. In the low photon flux regime, the incident-photon-ratio definition[51] is equivalent to the detected-photon-ratio definition of SBR used in existing literature[52,53] that deals with low-flux LiDAR imaging.

Figure 3 a shows simulation results under low $SBR_{total}$ regime ($SBR_{total} = 2/50$, $SBR_{pulse} = 8.2$), sub-signal photon regime ($SBR_{total} = 0.2/10$, $SBR_{pulse} = 4.1$), high background flux regime ($SBR_{total} = 10/2000$, $SBR_{pulse} = 1.0$), and outdoor conditions ($SBR_{total} = 3000/3000$, $SBR_{pulse} = 204.8$) with high background flux and long depth range. Under each illumination condition, we simulated the 3D photon transient cubes using the first photon captured in each laser cycle for realistic SPAD measurements (see "Methods" and Supplementary Table S1). For static, short-depth-range scenes we used the Middlebury dataset[54] (Fig. 3a rows 1–3). For long-depth-range scenes with motion, we used CARLA simulator[55] (Fig. 3a row 4). The measured photon transients are severely corrupted by noise and pileup as shown in Fig. 3b. We recover the photon fluxes by our method first and then estimate depths through traditional matched filtering (MF)[56]. We compare our results with matched filtering and two other state-of-the-art methods for photon-efficient LiDAR imaging: a statistical approach[40] and a learning-based approach[41]. These comparison methods directly estimate depths without any pre-processing of the photon transient cubes. The performance metric used is the percent of inlier pixels at different error thresholds of 0.2, 0.5, and 1%.

The statistical approach[40] relies on the intuition that signal photons cluster better than the background photons in the time domain. This assumption breaks down in the sub-photon regime where it is challenging to reliably locate signal photon clusters, and in the high background flux regime where spurious background photons may appear clustered. The learning-based approach[41] performs well in the low SBR setting for which it was trained, but fails under the out-of-distribution challenging flux regimes. Although its performance can be improved by fusion with additional intensity images, it is still

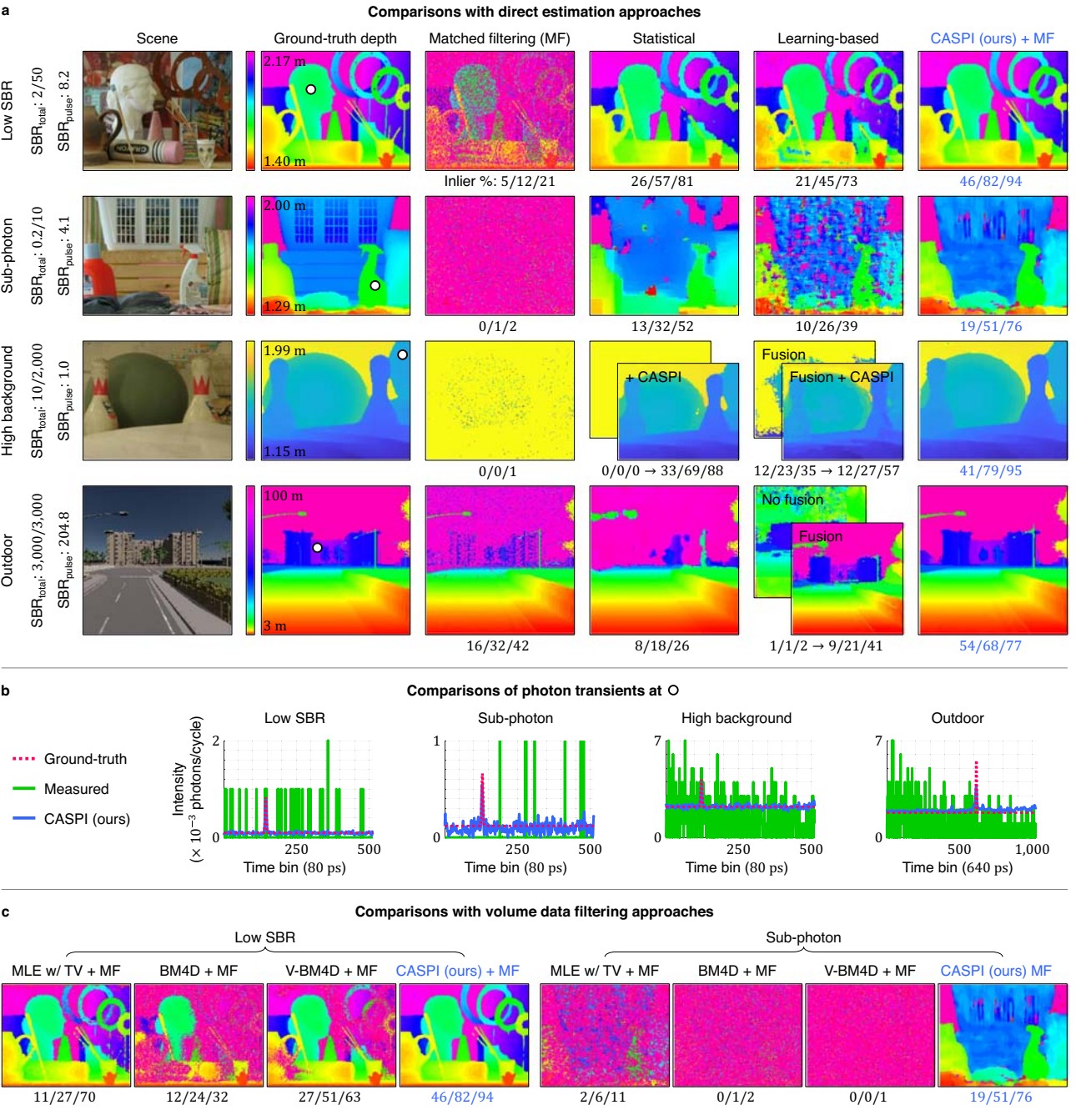

**Fig. 3 | CASPI for single-photon LiDAR. a** LiDAR imaging via CASPI is compared with matched filtering (MF)[56], a statistical approach[40], and a learning-based approach[41] in various illumination conditions using data from the Middlebury dataset and CARLA autonomous-driving simulator. The three numbers underneath each depth map show the percent fraction of inlier pixels that lie within 0.2, 0.5, and 1% of the true depths. As shown in the third row, CASPI can enhance the performance of existing methods. **b** CASPI recovers the latent photon fluxes over various flux regimes and enables reliable depth estimation even in challenging conditions. **c** State-of-the-art volumetric data denoising approaches fail to recover true photon fluxes in challenging flux regimes, which leads to unreliable depth estimates.

challenging to recover depth details in the Outdoor SBR regime (Fig. 3a row 4). In the Outdoor scene, even with a seemingly high $SBR_{pulse} = 204.8$, direct estimation approaches fail to reconstruct farther scene points such as the building. This is because the scene has $SBR_{total} = 3000/3000$; The large number of background photons causes strong pileup artifacts for farther scene points. Note that in the high background flux regimes, we applied Coates' correction[29] to all compared approaches to mitigate pileup. CASPI provides the best depth accuracy under all lighting conditions by reliably recovering the latent photon fluxes (Fig. 3b). Our method is complementary to

existing algorithms and can enhance their performance by providing the latent fluxes as shown in Fig. 3a row 3.

In Fig. 3c we compare our approach with various volumetric data denoising methods: maximum likelihood estimation with total-variation (TV)[57], BM4D for volumetric data denoising[48], and V-BM4D for video filtering[49]. Final depth values are estimated using matched filtering[56]. The compared approaches show some improvement in depth estimation in the low SBR regime, but struggle in the sub-photon regime due to the lack of reliable knowledge of noise statistics (see Supplementary Discussion). In Fig. 4 we show experimental results on

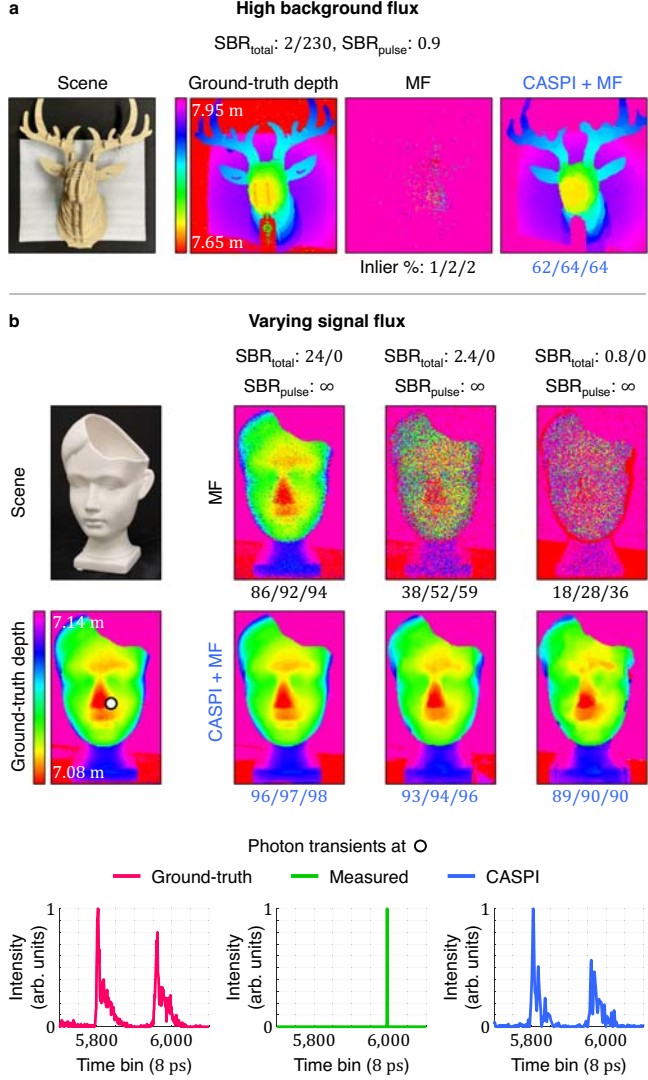

**a** **High background flux**

SBR$_{total}$: 2/230, SBR$_{pulse}$: 0.9

Scene | Ground-truth depth | MF | CASPI + MF

7.95 m
7.65 m

Inlier %: 1/2/2 | 62/64/64

**b** **Varying signal flux**

SBR$_{total}$: 24/0     SBR$_{total}$: 2.4/0     SBR$_{total}$: 0.8/0
SBR$_{pulse}$: ∞       SBR$_{pulse}$: ∞         SBR$_{pulse}$: ∞

Scene

MF

86/92/94     38/52/59     18/28/36

Ground-truth depth

7.14 m
7.08 m

CASPI + MF

96/97/98     93/94/96     89/90/90

Photon transients at ○

— Ground-truth    — Measured    — CASPI

Intensity (arb. units)   Time bin (8 ps)

**Fig. 4 | CASPI for single-photon LiDAR experiments. a, b** Our approach succeeds in recovering 3D geometry both in (**a**) high background flux and (**b**) sub-photon regimes where the conventional approaches fail. CASPI is robust to non-idealities in real-world experimental datasets (e.g., non-Gaussian bi-modal laser pulse shapes shown in the last row). This demonstrates the practical versatility of our approach across a wide range of operating conditions.

real-world data captured by a single-photon LiDAR hardware prototype (see "Methods" and Supplementary Table S2). These experiments have challenging operating conditions due to not just the lighting conditions but also a non-ideal bi-modal instrument response function (IRF) which deviates from a Gaussian IRF shape that is conventionally assumed. As shown in Fig. 4, conventional approaches only work in the high signal and low background flux regimes. In contrast, our method recovers high resolution 3D geometry over a wider range of illumination conditions including high background flux and sub-signal photon regimes.

**Recovering multipath transients**

Conventional LiDAR imaging assumes that sensor pixels only receive direct light components from the scene points. However, sensor pixels may receive indirect radiance due to multiple reflections or volumetric scattering, especially in flash-illumination-based LiDARs. This effect is called multipath interference[58]. On the one hand, multipath effects can cause large systematic depth errors for conventional LiDAR imaging,

while on the other hand, they can be exploited to recover 3D scene geometry out of the direct line-of-sight[14]. Since CASPI does not assume any prior constraints on the number of reflections or shapes of photon transients, it can be used to recover the entire time-varying photon flux waveform including multipath effects.

We simulated a photon transient cube (see Supplementary Table S1) with only $N_{sig} = 10$ signal photons/pixel for an indoor kitchen scene shown in Fig. 5. This scene contains many transients containing multipath effects due to the presence of numerous corners, and overall complex scene geometry. The ground-truth data[59] was generated using a physically accurate Monte Carlo flash LiDAR simulator that included multipath effects. Figure 5 shows the comparisons between measured transients, ground-truth transients, and transients recovered by CASPI at four different scene points. CASPI faithfully recovers not only the direct reflection, but also indirect multipath components. See Supplementary Results for additional LiDAR imaging results on intensity estimation (Fig. S2), depth estimation with high-quality intensity (Fig. S3), and depth estimation at different spatial resolutions (Fig. S4).

**Low photon count FLIM**

We validate the effectiveness of CASPI for FLIM in challenging low photon count datasets. Two FLIM datasets of fixed, labeled BPAE cells were collected with different acquisition times with average photon counts per pixel of 10 and 500, respectively. See Supplementary Table S4 for details on photon transient cube specification. The photon transient cube with 500 photons/pixel is used to obtain ground-truth lifetimes. We apply CASPI to the photon transient cube with 10 photons/pixel to recover the temporal fluorescence emission and estimate the lifetimes using maximum-likelihood estimation (MLE), one of the most widely used estimation techniques for FLIM analysis[60]. For comparisons, we enhance the SNR of the photon transient cube with 10 photons/pixel by 7 × 7 spatial binning (similar to the spatial size of the cubelet in CASPI) and estimate the lifetime for each pixel using MLE. Furthermore, we apply the BM3D[47] denoising technique to the lifetime image to reduce the estimation error. As shown in Fig. 6a, our results show substantially more reliable lifetime estimates than spatial binning + BM3D. CASPI achieves 5× better performance in terms of root-mean-square error (RMSE). Based on these reliable estimates, DAPI stained nuclei and mitotracker stained mitochondrial structures are separable, which is challenging with spatial binning + BM3D (Fig. 6a). Note that spatial binning + MLE is considered state-of-the-art and is available in FLIM data analysis software packages[60].

Next, we imaged living cells using their autofluorescence contrast in unlabeled live cells. The low yield of photons from intrinsic markers such as NADH and NADPH requires long acquisition times. To provide a viable, long-term imaging situation, we performed a time-lapse collection of FLIM datasets on living cells under a multi-photon excitation microscope (see "Methods"). These are temporal sequences of 3D photon transient cubes with rapid non-rigid motion. Figure 6b, c shows the lifetime estimates of epithelial cells under physiological conditions. After applying CASPI to the sequences of photon transient cubes, lifetime estimates are obtained using two different standard methods: MLE (Fig. 6b) and linear fitting on the log-transformed 1D photon transients (Fig. 6c). For comparisons, we apply spatial binning to the sequences of photon transient cubes and BM3D denoising to the lifetime images. As shown in Fig. 6b and c, the conventional method of spatial binning + BM3D denoising fails to recover the underlying photon transients. CASPI recovers detailed structural characteristics of these live samples by restoring the latent photon fluxes (last column) even in such low signal conditions and in the presence of motion. See Supplementary Video for detailed comparisons.

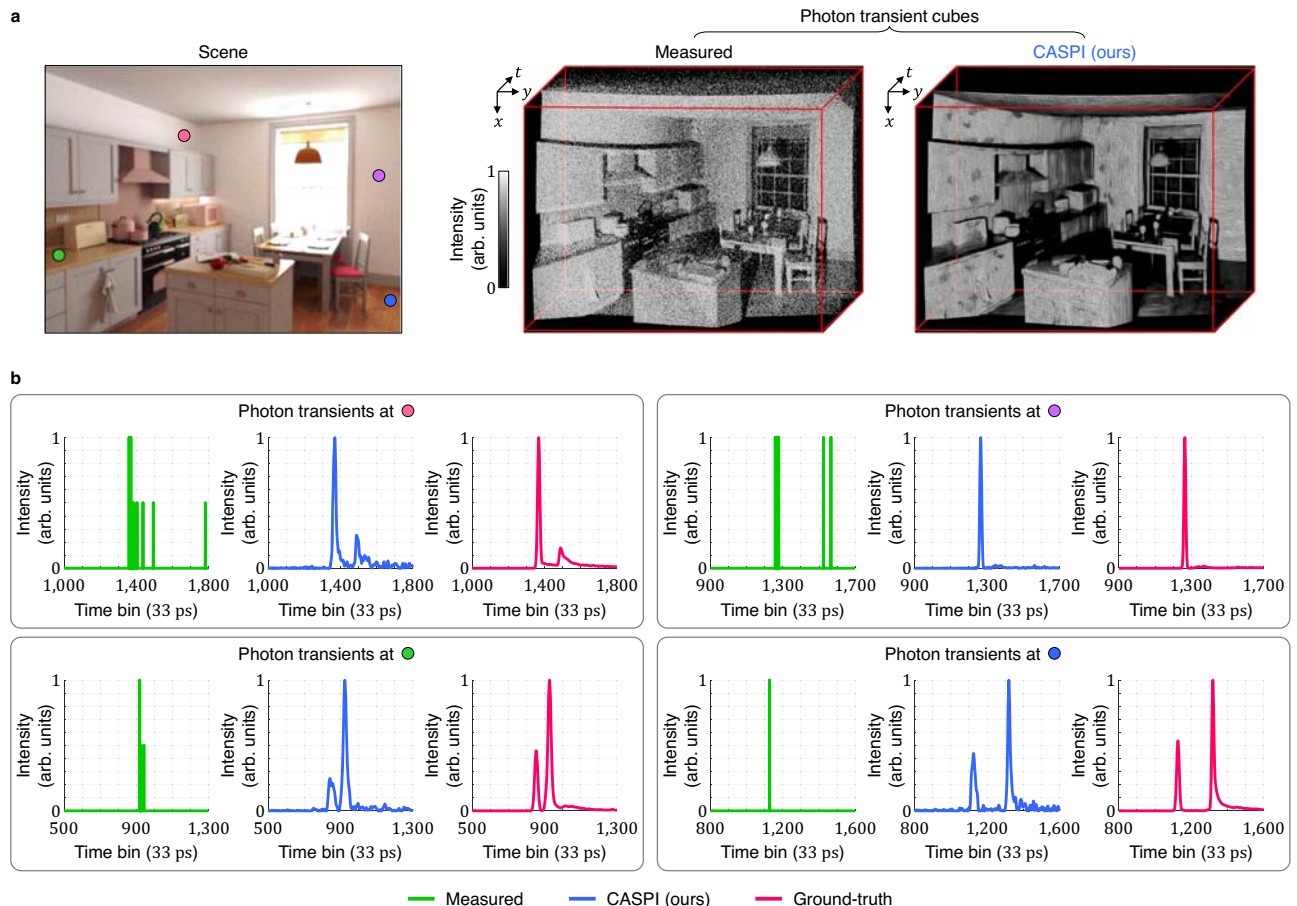

**Fig. 5 | Recovering multipath transients. a, b** The kitchen scene was simulated using a photo-realistic graphics rendering engine which emulates active single-photon imaging, and includes multipath effects that cause the ground-truth photon transients to deviate significantly from an ideal Gaussian pulse shape. CASPI recovers the entire temporal profiles of light transport, including indirect multipath reflections as well as direct reflections. **a** The top row shows the measured photon transient cube with as few as average 10 signal photons/pixel and the photon fluxes recovered by CASPI. **b** The second and third rows show accurate recovery of a variety of transients that include multiple peaks due to multipath effects.

## Comparisons with global analysis for multi-exponential decay in FLIM

In general, it is challenging to estimate the relative contributions and fluorescence lifetimes (especially in photon-starved regimes) for multi-exponential fluorescence lifetime decay models where the decay kinetics are given by a sum of two or more exponential functions. Pixel-wise fitting may be unreliable if fewer than 1000 photons are available per pixel[61]. Global analysis methods perform better in these low SNR scenarios by simultaneously analyzing all photon transients of the FLIM image[62]. Here we show that CASPI can recover bi-exponential transients at photon flux levels considerably lower than the conventional rule-of-thumb 1000 photons/pixel[61], while providing more reliable estimates than global fitting. Figure 7 shows the estimated relative contributions in a bi-exponential lifetime model using three different methods: pixel-wise fitting, global fitting[63], and pixel-wise fitting on photon transients recovered with CASPI. These results were obtained using the open-source FLIMfit software package[64]. The photon transient cube (see Supplementary Table S3 for details) was simulated using a fixed 200 signal photon counts per pixel, two different lifetimes $\tau_1 = 3$ ns and $\tau_2 = 1.5$ ns (fixed for the entire FLIM image), and two relative contributions $\beta_1$ and $\beta_2 (= 1 - \beta_1)$ that vary over the image (ground-truth shown in Fig. 7a). Due to the low number of photons, pixel-wise fitting suffers from strong Poisson noise artifacts (Fig. 7b). Although global fitting provides some improvement (Fig. 7c), CASPI provides the most reliable estimates of $\beta_1$ and $\beta_2$ (5× better performance in terms of RMSE as shown in Fig. 7d). It is also worth noting that global analysis techniques assume that the lifetimes are constant over the entire image and only their relative contributions vary on a per-pixel basis. CASPI does not assume such a prior making it applicable in more general settings where both the lifetimes and the relative contributions vary over the FLIM image. See Supplementary Results for additional FLIM results with low signal photon counts (Fig. S5), lifetime estimation accuracy (Fig. S6), and lifetime estimation with the Z-stack data (spatial sequences of photon transient cubes) (Fig. S7).

## Discussion

Since CASPI employs cubelet-based transforms, processing each cubelet sequentially can result in long compute times (about 20 min with a $256 \times 256 \times 256$ photon transient cube by unoptimized MATLAB implementation). Fortunately, each cubelet can be processed independently making our method amenable to massively parallel processing (e.g., on low-power GPUs) enabling real-time implementations on future hardware chips. Further speed-ups can be obtained by implementing fast Fourier transforms in hardware or computing them optically. Furthermore, considering that the temporal locations of the signal features also convey important information in specific applications (e.g., LiDAR), wavelet transforms may achieve better performance than Fourier transforms in recovering photon fluxes.

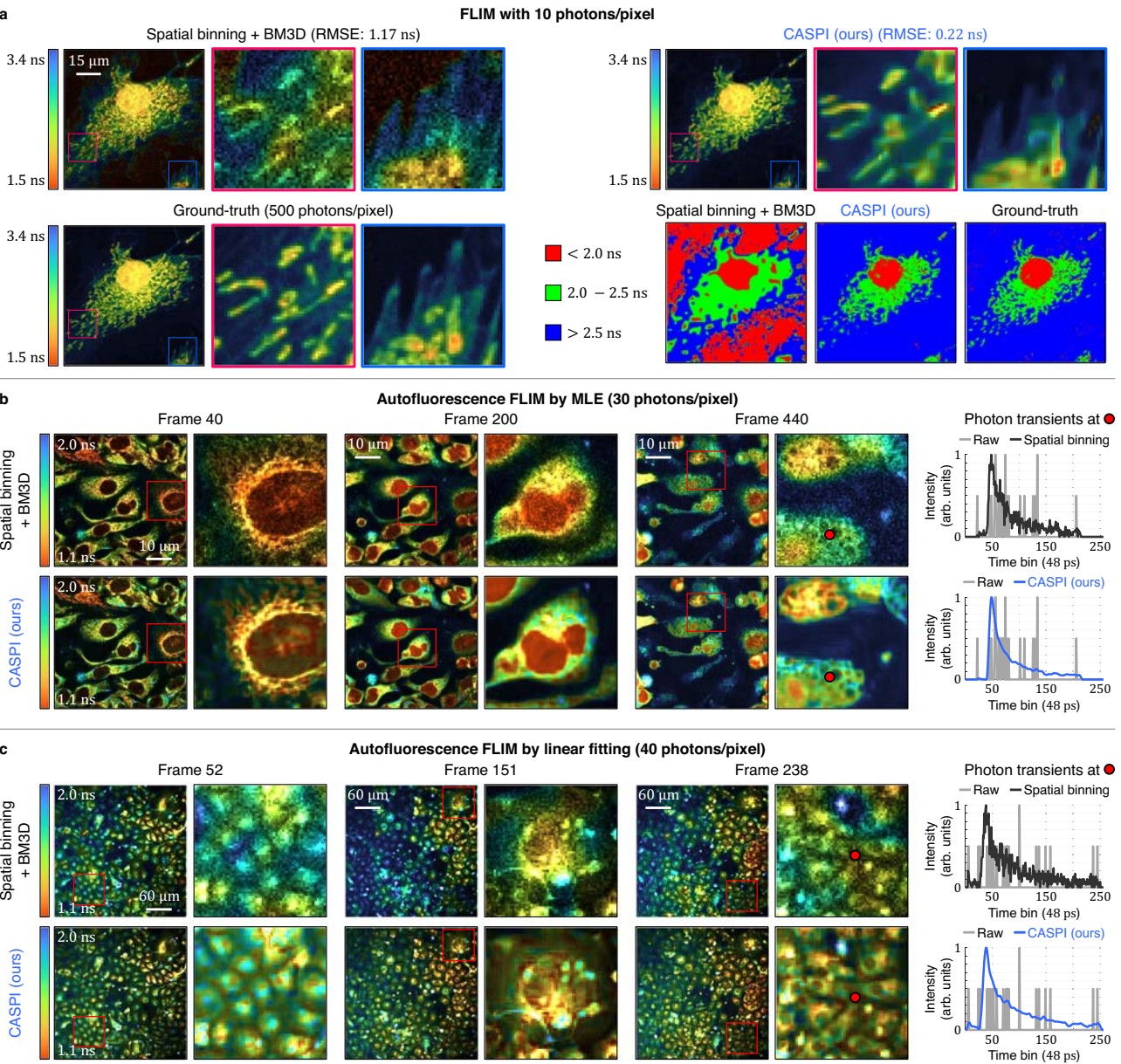

**Fig. 6 | CASPI for FLIM. a** CASPI enables reliable lifetime estimates with as few as 10 photons per pixel and achieves 5× better performance in root-mean-square error (RMSE) compared to spatial binning of the photon transient cubes followed by BM3D applied to the lifetime estimates. The sample imaged here contains fixed BPAE endothelial cells with fluorescent labels. DAPI stained nuclei and mitotracker stained mitochondrial structures are separable using CASPI even with 50× fewer photons than the ground-truth. **b, c** CASPI recovers the underlying transients from the autofluorescence emission (last column) from the low photon count datasets of autofluorescence of biological samples. When combined with existing fitting methods ((**b**) state-of-the-art maximum likelihood estimation (MLE), or (**c**) naive linear-fit on log-transformed histograms), CASPI enables to recover fine structures and details even for moving living cells (see also, Supplementary Video).

Some limitations of SPCs may get resolved with future hardware improvements. For example, pileup can be addressed by faster sampling mechanisms such as multi-event time-to-digital converters (TDCs), circuits with shorter dead-times, and multi-bit gated pixels. The improvements due to CASPI are complementary to hardware innovation and can be used to resolve the limitations of not only first photon detection in photon-flooded regime, but also low photon counts in photon-starved regime when there are no/minimal pileup distortions. Low SNR caused by low photon counts is a fundamental problem often encountered in many real-world imaging applications—imagine a LiDAR capturing a low-albedo object at a long distance, or a FLIM imaging scenario that is constrained to low laser power and low capture time to avoid photobleaching. Thanks to its versatile training-

free and blind operation, we envision CASPI becoming an integral part of various active SPC data pipelines. In 3D imaging, it may enable long-range low-power flash LiDARs for future autonomous vehicles and robotics applications. CASPI will also enable real-time in vivo observation of fluorescence lifetime contrasts in biomedical imaging applications to assess metabolism or systemic changes due to cellular activity.

## Methods
### Image formation for LiDAR and FLIM
An active imaging system consists of a laser source that emits short light pulses into the scene and a sensor that captures the returning photons as shown in Fig. 1a. Let $s(t)$ denote the shape of the laser pulse

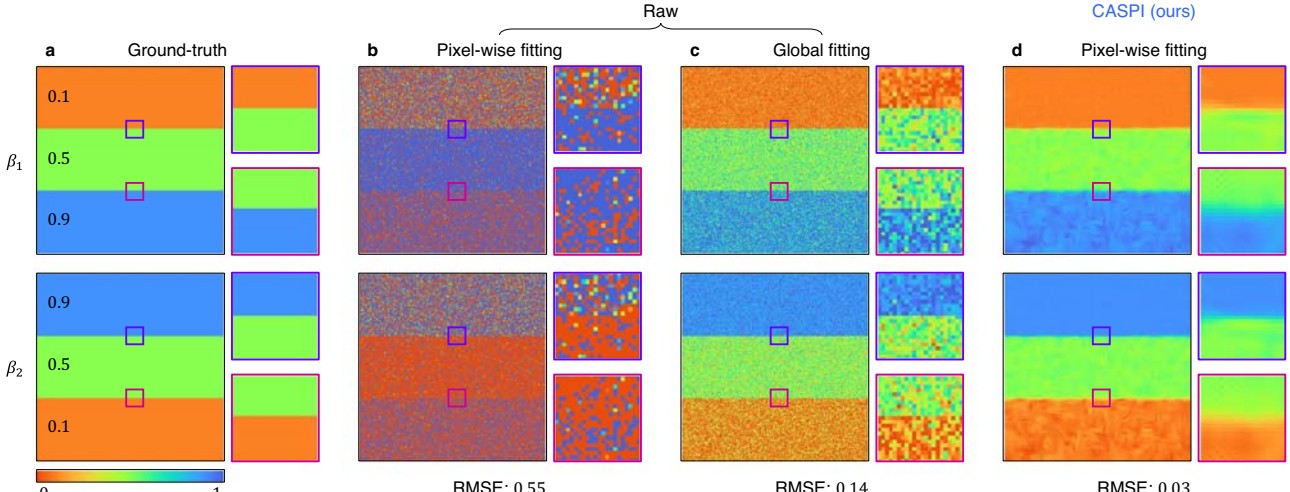

**Fig. 7 | Comparisons with global analysis for bi-exponential decay in FLIM.**
**a** The ground-truth used for simulating photon transient cubes consists of two invariant lifetimes $\tau_1 = 3$ ns and $\tau_2 = 1.5$ ns with relative contributions $\beta_1 = 0.1, 0.5, 0.9$ and $\beta_2 = 1 - \beta_1$ that vary over the field-of-view. **b** When estimating the parameters of a multi-exponential decay model, pixel-wise fitting is often unreliable if fewer than 1000 photons are available per pixel. **c** Global analysis provides better estimation

accuracy than pixel-wise fitting by considering all photon transients simultaneously assuming that the lifetimes are spatially invariant. **d** CASPI can reliably estimate the parameters of a bi-exponential decay model in FLIM even with as few as 200 photons per pixel without the spatial-invariance assumption that global analysis relies on. After applying CASPI, even a pixel-wise fitting provides better estimates than global analysis by 5× in RMSE.

(normalized to unit sum), and $h(t)$ be the scene response function. The photon flux, $\Phi(t)$ reaching the sensor is:

$$\Phi(t) = \Phi_{\text{sig}} s(t) * h(t) + \Phi_{\text{bkg}}, \tag{1}$$

where $\Phi_{\text{sig}}$ is the signal component of the incident flux which encapsulates the laser source power along with scene-dependent factors such as distance-squared fall-off, scene reflectivity and BRDF; $\Phi_{\text{bkg}}$ is the background component accounting for ambient light and spurious photon events recorded by the sensor due to dark noise; and * denotes the convolution operation. Using different scene response functions, $h(t)$, Eq. (1) can be used to mathematically model the data capture process in a wide range of active single-photon imaging applications. In this work we focus on LiDAR and FLIM.

For LiDAR imaging, the scene depth is encoded in the round-trip time-of-flight of the laser pulse with the assumption that the laser source and the sensor are co-located. Thus, the scene response for LiDAR imaging is modeled as:

$$h_{\text{LiDAR}}(t; d) = \delta\left(t - \frac{2d}{c}\right), \tag{2}$$

where $\delta(t)$ denotes the Dirac delta function; $d$ is a true (unknown) depth of the scene point; and $c$ is the speed of light. The image formation model for a LiDAR can be obtained by replacing $h(t)$ in Eq. (1) with $h_{\text{LiDAR}}(t; d)$.

For FLIM, the fluorescence lifetime of the material/molecule is defined as the decay rate of the exponentially decaying fluorescence emission intensity after excitation. Assuming a single exponential decay model, the fluorescence transient response is given as:

$$h_{\text{FLIM}}(t; \tau) = \frac{1}{\tau} e^{-\frac{t}{\tau}} \quad (t \geq 0), \tag{3}$$

where $\tau$ is the fluorescence lifetime. Here $h_{\text{FLIM}}(t; \tau)$ is normalized so that $\int_0^\infty h_{\text{FLIM}}(t; \tau) \, dt = 1$. The image formation model for FLIM is obtained by substituting $h(t) = h_{\text{FLIM}}(t; \tau)$ in Eq. (1).

## SPAD histogram formation

In this section we derive an image formation model for the transient histograms captured by a SPAD-based SPC. SPAD-based active imaging relies on the time-correlated single-photon counting (TCSPC) principle[65]. A scene point is illuminated repeatedly by a periodic train of laser pulses. In each laser cycle (a period between the laser pulses), the SPAD detects only the first returning photon, following which it enters a dead-time (~100 ns). During the dead-time intervals, the SPAD pixel cannot detect any additional photons. The arrival time of the first incident photon with respect to the start of the most recent laser cycle is recorded, and this is repeated over many laser cycles to build the histogram of photon detection times.

Because the SPAD has a finite time resolution (tens of ps), we consider a time-discrete version of the photon flux reaching the SPAD. From Eq. (1),

$$\Phi[n] = \int_{(n-1)\Delta_t}^{n\Delta_t} \Phi(t) \, dt \quad (n \in \{1, 2, \ldots, N_t\}), \tag{4}$$

where $n$ is the time bin index, $\Delta_t$ is the time bin size and $N_t$ is the number of time bins in the histogram. The incident photon counts at the $n^{\text{th}}$ time bin follow a Poisson distribution with the mean given by Eq. (4). The probability that at least one photon is incident during the $n^{\text{th}}$ time bin is given by:

$$q[n] = 1 - e^{-\Phi[n]}. \tag{5}$$

Thus, the probability of detecting a photon in the $n^{\text{th}}$ time bin is:

$$p[n] = q[n] \prod_{i=1}^{n-1} (1 - q[i]). \tag{6}$$

In a low flux regime where $\Phi[n] \ll 1 \, \forall \, n$, $p[n] \propto \Phi[n]$ and the SPAD histogram approximates the incident photon flux well with high number of laser cycles. However, if the incident flux is too high, there is a high probability that >1 photon is incident on the SPAD pixel in the same laser cycle. The captured histogram suffers from pileup distortions because the SPAD captures only the first returning photon for each laser cycle.

## Correcting pileup distortion

In a high flux regime, the relationship $p[n] \propto \Phi[n]$ does not hold, and the histogram gets skewed towards earlier time bins as shown in Fig. 1b. This non-linear distortion is called photon pileup. Theoretically, it is possible to invert the pileup distortion computationally using the Coates' correction[29]:

$$\widehat{\Phi}[n] = \ln\left(\frac{N_{\text{cycle}} - \sum_{i=1}^{n-1} H[i]}{N_{\text{cycle}} - \sum_{i=1}^{n-1} H[i] - H[n]}\right), \qquad (7)$$

where $\widehat{\Phi}[n]$ is the estimate of the incident flux at the $n^{\text{th}}$ time bin, $N_{\text{cycle}}$ is the total number of laser cycles, and $H[i]$ is the photon counts at the $i^{\text{th}}$ time bin. Although the Coates' correction reduces the pileup distortion to some extent, it has the detrimental effect of amplifying noise at later bins as shown in Fig. 1b. Pileup distortion can also be mitigated through hardware improvements that allow shorter SPAD dead-times and faster timestamp processing using multi-hit TCSPC modules. We expect CASPI will play a complementary role to existing and upcoming computational and hardware approaches to deal with pileup distortions.

## CASPI: algorithm details

The raw data captured from the SPC consists of a stream of photon timestamps at each pixel location. For each camera pixel, we construct a 1D photon transient ($\in \mathbb{Z}_+^{N_t}$, where $N_t$ is the number of time bins), a histogram of photon counts as a function of photon detection time. By repeating this for all pixels, we can build a 3D photon transient cube ($\in \mathbb{Z}_+^{N_y \times N_x \times N_t}$, where $N_y$ and $N_x$ are the numbers of rows and columns in the SPC pixel array) as shown in Fig. 2. CASPI requires this photon transient cube and the laser source prior as inputs. Optionally, a high-quality 2D intensity image ($\in \mathbb{R}_+^{N_y \times N_x}$) capturing the same scene as the SPC can be used as additional input when available from another co-located image sensor (e.g., a low-light RGB camera). Figure 8a shows an overview of the proposed algorithm. First, we apply Coates' correction[29] to each 1D photon transient to reduce any pileup distortion present in the 3D photon transient cube. Under severe pileup conditions, Coates' correction may amplify noise in the photon transient cube (the more severe the distortion is, the higher the noise will be). CASPI ameliorates this noise during the next photon flux recovery stage. The photon fluxes are recovered in a hierarchical manner. We recover the photon fluxes by first using only local spatio-temporal correlations to find similar cubelets. After collecting the similar cubelets, we recover the final photon fluxes using both local and non-local correlations collaboratively.

Figure 8b shows the steps for recovering the photon fluxes from the noisy photon transient cube in the hierarchical processing. We estimate the noise and initial fluxes first, and the refined flux estimates are obtained by Wiener filtering which is known to be optimal in a mean-squared-error sense[50]. Figure 8c shows the procedure for the noise and initial flux estimation. A 3D photon transient cubelet ($\in \mathbb{R}_+^{C_y \times C_x \times N_t}$, where $C_y$ and $C_x$ are the numbers of rows and columns of the cubelet, respectively) is defined for each pixel such that the pixel is located at the upper left front corner of the cubelet. A cubelet is the fundamental data structure used for recovering the fluxes using local correlations. A 4D photon transient set (a set of similar cubelets $\in \mathbb{R}_+^{C_y \times C_x \times N_t \times N_{\text{sim}}}$, where $N_{\text{sim}}$ is the number of similar cubelets) is the fundamental data structure used when we exploit local and non-local correlations. We used $C_x = 8$, $C_y = 8$, and $N_{\text{sim}} = 10$ in our simulations and experiments except for Fig. 6a, where $C_x = C_y = 4$ were used for better recovery of the small mitochondrial structures.

In the Fourier domain of the 3D photon transient cubelet (or 4D photon transient set if non-local correlations are also available), the pure noise band is isolated based on the laser source prior, where the noise amount is estimated accurately (details of noise band estimation are

given in the next section). To get the initial flux estimates, thresholding and guided photon processing are selected adaptively depending on the SNR of the 3D photon transient cubelet (or 4D photon transient set if non-local correlations are also available). This SNR is defined as:

$$R = \frac{\mathbb{E}\left[|B_{\text{noise}}^c|^2\right]}{\mathbb{E}\left[|B_{\text{noise}}|^2\right]}, \qquad (8)$$

where $|B_{\text{noise}}|^2$ is the energy of the Fourier coefficients inside the pure noise band, $|B_{\text{noise}}^c|^2$ is the energy outside the pure noise band, and $\mathbb{E}[\cdot]$ denotes the expected value (mean). Thresholding is selected if $R > R_{th}$, and guided photon processing is selected otherwise. $R_{th} = \frac{1}{0.8}$ was used for initial flux estimation using only local correlations and $R_{th} = \frac{1}{0.9}$ for initial flux estimation using local and non-local correlations. If thresholding is selected, initial flux estimates are obtained by thresholding with the noise threshold in the Fourier domain. We define the noise threshold, $\delta_{\text{noise}}$ as a statistical upper bound for the magnitude of the pure noise band (Fig. 2a):

$$\delta_{\text{noise}} = \left(1 + 4\sqrt{\frac{4}{\pi} - 1}\right) \mathbb{E}[|B_{\text{noise}}|]. \qquad (9)$$

See Supplementary Derivation for the derivation of $\delta_{\text{noise}}$.

If the initial flux estimation is performed on the 4D photon transient set using local and non-local correlations, all recovered 3D photon transient cubelets return to their original locations. After the initial flux estimation is repeated for all pixels, multiple initial flux estimates are produced for each pixel location due to overlapping regions between different photon transient cubelets. A single initial flux estimate for each pixel is obtained by computing the weighted average of these multiple initial flux estimates:

$$\widetilde{\Phi} = \frac{\sum_{i=1}^{N_l} \omega_i \widetilde{\Phi}_i}{\sum_{i=1}^{N_l} \omega_i}, \qquad (10)$$

where $N_l$ is the number of all overlapping cubelets on the pixel, $\widetilde{\Phi}_i$ is the initial flux estimate for the pixel by the $i^{\text{th}}$ overlapping cubelet, and $\omega_i$ is the weight assigned to the $i^{\text{th}}$ overlapping cubelet. The weights are in inverse proportion to the noise components of each cubelet:

$$\omega_i = \frac{1}{\mathbb{E}\left[|B_{\text{noise},i}|^2\right]}, \qquad (11)$$

where $|B_{\text{noise},i}|^2$ is the energy of the Fourier coefficients inside the pure noise band of the $i^{\text{th}}$ overlapping cubelet.

Based on the initial flux and the noise estimates, we apply Wiener filtering to obtain the final flux estimates as shown in Fig. 8d. Wiener filtering attenuates the Fourier coefficients of the noisy 3D photon transient cubelet (when using only local correlations) or noisy 4D photon transient set (when using local and non-local correlations) by element-wise multiplication with the Wiener coefficient:

$$W = \frac{|A|^2}{|A|^2 + \mathbb{E}\left[|B_{\text{noise}}|^2\right]}, \qquad (12)$$

where $|A|^2$ is the energy of the Fourier coefficients of the initial flux estimates, and $|B_{\text{noise}}|^2$ is the energy of the Fourier coefficients inside the pure noise band. After the Wiener filtering is repeated for all pixels, multiple flux estimates are produced for each pixel as with the initial flux estimation. A single final flux estimate for each pixel is obtained by a weighted average of the multiple Wiener-filtered flux estimates using Eqs. (10) and (11) where $\widetilde{\Phi}_i$ is now the Wiener-filtered flux estimate instead of the initial flux estimate in this case.

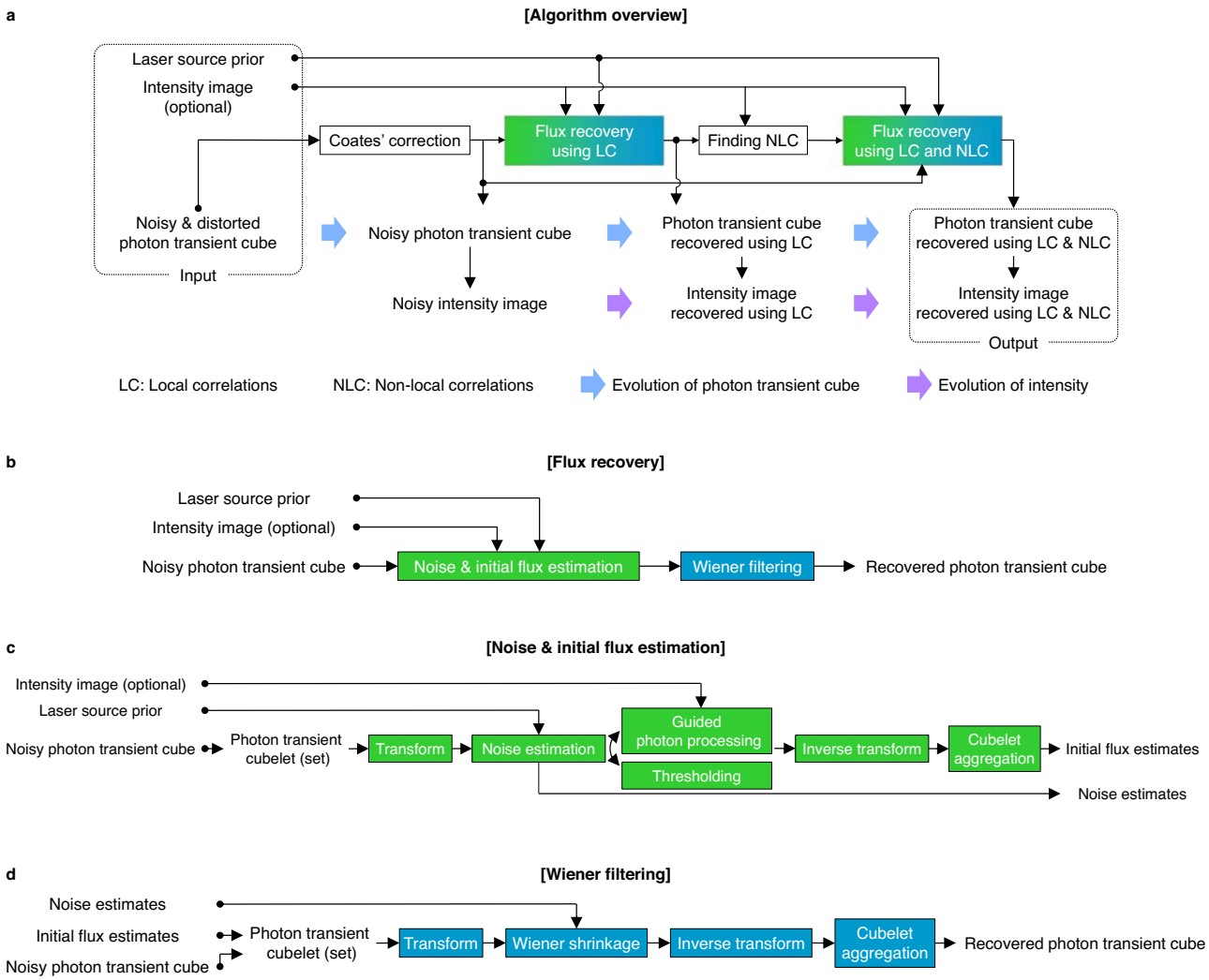

**Fig. 8 | Algorithm of CASPI. a** A noisy and distorted photon transient cube, laser source prior, and a high-quality intensity image (optional) are supplied as input to our algorithm. After reducing the potential pileup distortions in the photon transient cube using Coates' correction, we recover the photon fluxes using local correlations (LC) to find similar photon cubelets. After collecting the similar cubelets, the final photon fluxes are recovered by exploiting local and non-local correlations (LC and NLC) collaboratively. **b** The photon fluxes are recovered by two steps. **c** The noise and initial fluxes are recovered first. **d** Flux estimates are further refined through Wiener filtering based on the estimated SNR.

## Isolating pure noise band

We use Gaussian laser pulses with FWHM (full-width at half-maximum) of 400 ps (Fig. 3a rows 1–3, Figs. 3c, 5, and 7) and 3400 ps (Fig. 3a row 4) for our LiDAR and FLIM simulations. In case of the Gaussian-shaped pulses, we define the pure noise band $B_{noise}$ as the band of frequencies above three standard deviations of the Gaussian spectrum as shown in Fig. 9a:

$$B_{noise} = \left\{ f | f > \frac{3\sqrt{2\ln 2}}{\pi\, \text{FWHM}} \right\}. \tag{13}$$

For LiDAR experiments, we use a non-Gaussian laser pulse with two asymmetric peaks (Fig. 9b) as measured during calibration of our hardware setup. Even for the non-Gaussian pulses, most of the signal energy is concentrated in the low-frequency band and the pure noise can be isolated as shown in Fig. 9b. We define the pure noise band for the non-Gaussian pulse used in our LiDAR experiments as all the

Fourier frequency bins where the magnitude is <1% of the maximum:

$$B_{noise} = \left\{ f | f > f_N \quad \text{and} \quad |\mathcal{L}(f_N)| = 0.01 \max_f |\mathcal{L}(f)| \right\}, \tag{14}$$

where $|\mathcal{L}(f)|$ is the Fourier magnitude of the instrument response function (IRF) at the frequency of $f$. For FLIM experiments, the pure noise band can be defined similarly from the IRF of the FLIM system.

## Finding non-local correlations

CASPI relies on finding similar cubelets to exploit both local and non-local correlations. To find the similar cubelets efficiently, the search space is defined on the 2D intensity image instead of the 3D photon transient cube. If a high-quality intensity image is available as additional input from another co-located imaging modality, it can be used to define the search space. Otherwise, we obtain a pseudo-intensity image by summing over the time dimension of the photon transient cube recovered using local correlations. For each pixel, we define a reference patch $P_R$ ($\in \mathbb{R}_+^{C_y \times C_x}$) on the intensity image such that the pixel is located at the upper left corner of the reference patch (in synchronization with the cubelet). Next, a $S_{intra} \times S_{intra}$ search window

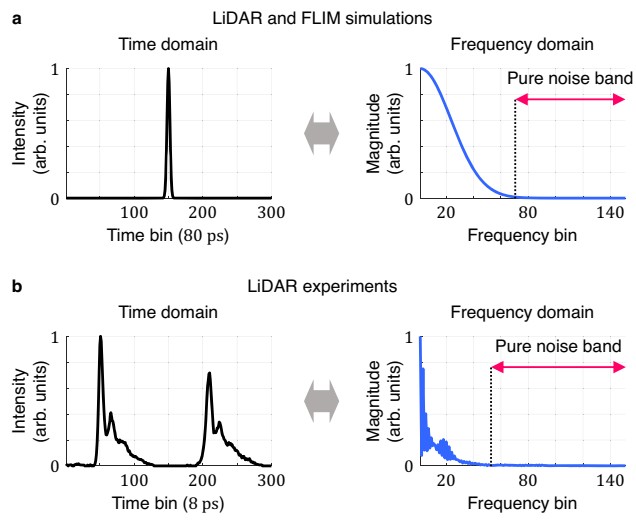

**Fig. 9 | Temporal laser profiles and corresponding pure noise bands. a, b (a)** Gaussian temporal laser profiles used for our LiDAR and FLIM simulations and **(b)** non-Gaussian temporal laser profiles used for our LiDAR experiments. In both cases, we can define the pure noise band by computing the Fourier spectral energy outside the main signal peak.

is centered at the reference patch, and a target patch $P_T$ ($\in \mathbb{R}_+^{C_y \times C_x}$) slides over the search window to find the similar image patches. We define a distance $d_{patch}$ between $P_R$ and $P_T$ as:

$$d_{patch} = \| P_R - P_T \|_2^2. \tag{15}$$

The set of similar image patches is defined as $N_{sim}$ number of image patches with the smallest $d_{patch}$ values. The locations of the similar cubelets are defined as the locations of the similar image patches. We used $S_{intra} = 21$ in our simulations and experiments.

**Guided photon processing**
Given the noisy photon transient cubelet ($\in \mathbb{R}_+^{C_y \times C_x \times N_t}$), the pseudo-intensity patch ($\in \mathbb{R}_+^{C_y \times C_x}$) is formed by summing over the time dimension. If a high-quality intensity image is available as an alternative input from another imaging modality, it can be used as the pseudo-intensity information. We normalize each pseudo-intensity patch such that the sum of the pixel values is 1. Let $\mathcal{P}$ ($\in \mathbb{C}^{C_y \times C_x}$) and $\mathcal{C}$ ($\in \mathbb{C}^{C_y \times C_x \times N_t}$) be the Fourier coefficients of the normalized pseudo-intensity patch and the noisy cubelet, respectively. We create $\mathcal{P}'$ ($\in \mathbb{C}^{C_y \times C_x \times N_t}$) by stacking $\mathcal{P}$ along the third dimension $N_t$ times. The guided photon processing is performed in the Fourier domain by element-wise multiplication of $\mathcal{P}'$ and $\mathcal{C}$ as shown in Fig. 10a. Figure 10b shows depth estimation results with initial flux estimates by thresholding only, guided photon processing only, and adaptive processing between thresholding and guided photon processing (depth values are estimated by applying matched filtering to the initial flux estimates). Adaptive processing produces the optimal initial flux estimates.

**Handling 4D photon transient sequences**
If multiple 3D photon transient cubes are available at different spatial or temporal positions, we can use a 4D photon transient sequence ($\in \mathbb{Z}_+^{N_y \times N_x \times N_t \times N_c}$, where $N_c$ denotes the number of the cubes) as the input of CASPI. Optionally, high-quality 3D intensity video ($\in \mathbb{R}_+^{N_y \times N_x \times N_c}$) can be used as additional input. The basic algorithm is the same as when the 3D photon transient cube is the input except that the search space centered at the reference patch is a 3D volume with a

dimension of $S_{intra} \times S_{intra} \times S_{inter}$ instead of $S_{intra} \times S_{intra}$. We use $S_{inter} = 11$ for our simulations and experiments.

**SPAD histograms for LiDAR simulations**
We build the SPAD histograms for LiDAR simulations as follows. The illumination conditions are quantified using the ratio $\left(N_{sig}/N_{bkg}\right)$, where $N_{sig}$ and $N_{bkg}$ are the average incident signal and background photon counts per pixel during the total laser cycles, respectively. The mean signal photon counts incident at pixel **p** in each cycle is given by:

$$\overline{N_{sig}}(\mathbf{p}) = \frac{N_{sig}}{N_{cycle}} \frac{I(\mathbf{p})}{D^2(\mathbf{p})} \mathbb{E}\left[\frac{D^2}{I}\right], \tag{16}$$

where $N_{cycle}$ is the total number of laser cycles, $I(\mathbf{p})$ is ground-truth intensity at **p**, $D(\mathbf{p})$ is ground-truth depth at **p**, and $\mathbb{E}\left[\frac{D^2}{I}\right]$ is the mean of pixel-wise division of the depth map squared by the intensity image. The mean background photon counts incident at **p** per cycle per time bin is given as:

$$\overline{N_{bkg}}(\mathbf{p}) = \frac{N_{bkg}}{N_{cycle} N_t} \frac{I(\mathbf{p})}{\mathbb{E}[I]}, \tag{17}$$

where $\mathbb{E}[I]$ is the mean of the intensity image. Note from Eqs. (16) and (17) that both signal and background fluxes are proportional to the intensity, and only the signal fluxes are inversely proportional to the square of the depth while the background fluxes remain constant regardless of the depth.

Assuming a Gaussian laser pulse, the time-discrete version of the signal flux incident at **p** is given by:

$$\Phi_{sig}(\mathbf{p}; n) = \overline{N_{sig}}(\mathbf{p}) \mathcal{N}\left(m = round\left(\frac{2d}{c\Delta_t}\right), \sigma = \frac{FWHM}{2\sqrt{2\log 2}\Delta_t}\right) \quad (n \in \{1,...,N_t\}), \tag{18}$$

where $\mathcal{N}$ is the normalized time-discrete Gaussian function with the mean $m$ and the standard deviation $\sigma$; $d$ is the depth; $c$ is the light speed and $\Delta_t$ is the time bin size. Note that $\sum_{n=1}^{N_t} \Phi_{sig}(\mathbf{p}; n) = \overline{N_{sig}}(\mathbf{p})$. The time-discrete version of the background flux incident at **p** is given by:

$$\Phi_{bkg}(\mathbf{p}; n) = \overline{N_{bkg}}(\mathbf{p}) \quad (n \in \{1,...,N_t\}). \tag{19}$$

Thus, the time-discrete version of the total flux incident at **p** is given by:

$$\Phi(\mathbf{p}; n) = \Phi_{sig}(\mathbf{p}; n) + \Phi_{bkg}(\mathbf{p}; n) \quad (n \in \{1,...,N_t\}). \tag{20}$$

In each laser cycle, we generate random photon counts according to Poisson statistics with $\Phi(\mathbf{p}; n)$ as the mean, and record the time bin index for the first incident photon (we assume quantum efficiency $\approx 1$ for simplicity). This is repeated over $N_{cycle}$ number of laser cycles to construct the SPAD histogram. Supplementary Table S1 shows the parameter values used to construct the photon transient cube (SPAD histograms over a 2D spatial grid) for LiDAR simulations.

**LiDAR experimental setup and data**
Our experimental LiDAR data consists of two datasets captured using the asynchronous single-photon imaging technique[24]. The datasets were obtained from the hardware prototype consisting of a 405 nm pulsed laser (Picoquant LDH-P-C-405B), a TCSPC module (Picoquant HydraHarp 400) and a fast-gated SPAD[66]. The laser was operated at a repetition frequency of 10 MHz for an unambiguous depth range of 15 m. Each dataset has a ground-truth photon transient cube acquired with long acquisition time without ambient light. For the face scene (Fig. 4b), we down-sampled the ground-truth data such that the

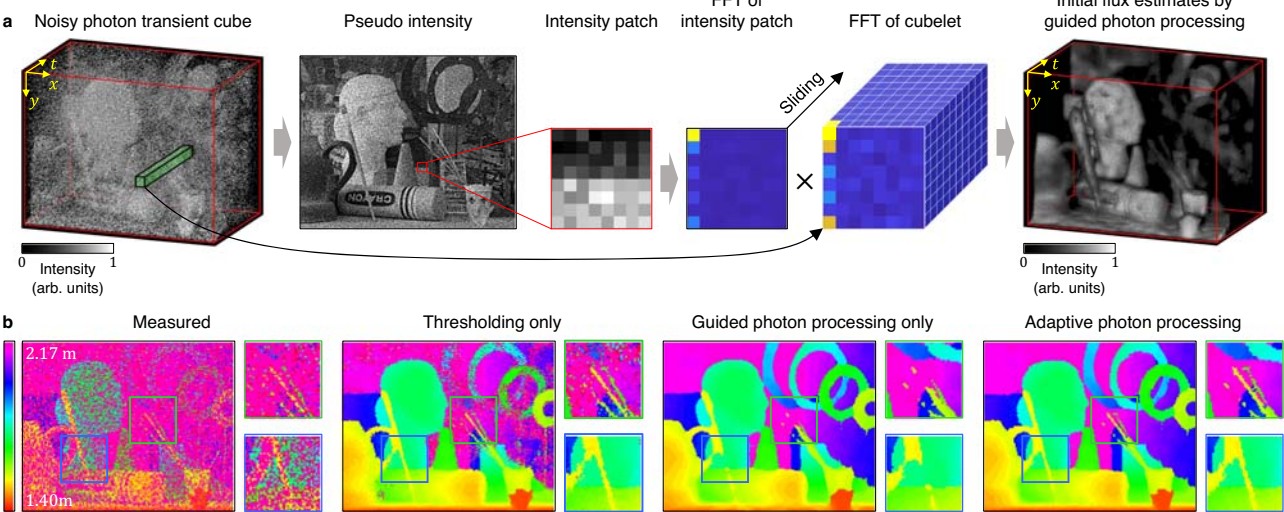

**Fig. 10 | Guided photon processing. a** We propose guided photon processing to recover the photon fluxes in low SNR scenarios. Guided photon processing uses spatial frequency correlations between the intensity image and the photon transient cube. **b** Guided photon processing and thresholding are adaptively selected based on the estimated cubelet SNR to generate the optimal initial flux estimates leading to optimal scene property estimates.

average signal photon counts per pixel are 24, 2.4, and 0.8. The deer scene (Fig. 4a) was captured under strong ambient illumination (>20,000 lux) high enough to cause pileup distortion. See Table S2 for more detailed data specifications.

### FLIM experimental setup and data

The FLIM data were acquired using two custom multiphoton microscopes. These microscopes use pulsed femtosecond lasers operating at a repetition rate $8 \times 10^7$ and 720 nm dichroic cut-off filter for separating fluorescence. The microscopes are designed for fluorescence lifetime imaging, and the FLIM data were collected using time-correlated single-photon counting (TCSPC) electronics[67,68]. The photons were collected using a photosensitive GaAsP PMT (H7422, Hamamatsu), and single-photon timings were determined by the SPC-150 timing module (Becker-Hickl GmbH, Berlin). Using the photodetector signal, galvanometer clocks, and pulsed laser sync signals, the photon arrival time is measured and single-pixel histograms are generated by TCSPC electronics. To allow photon counting electronics to operate at full capacity, the detector was set to operate at a constant gain. To perform the scanning and record the single-pixel histograms, we used two of our lab-developed laser scanning microscopy (LSM) tools, OpenScan (v0.D2020.03.11) and WiscScan (v7.5). To increase the number of frames used in a single 3D cube, we increased the collection time per FLIM dataset in the BH-150 parameters.

All cells were grown at (37 °C, and 5%) $CO_2$ in Dulbecco's modified Eagle's media with 10% fetal bovine serum. The cells were plated onto MatTek 35-mm glass-bottom dishes for imaging. The live-cell imaging was carried out using an imaging incubator that maintains humidity, temperature, and $CO_2$ levels at physiological conditions best suited for that cell line (37 °C, >90% RH and 5%). MCF10A epithelial cells (ATCC, CRL-10317) were unlabeled and provided as a gift from the Ponik Lab, UW-Madison and use a modified culture media, supplemented with 5% Horse Serum, 10 µg/ml of bovine insulin, 500 ng/ml of hydrocortisone, and 20 ng/ml of epithelial growth factor[68]. For the mCherry-labeled HeLa cells, transfection of HeLa cells (ATCC, CCL2) with H2B-mCherry (20972, Addgene) plasmid was performed using Lipofectamine 2000. The transfected cells were frozen in DMSO and stored; later the frozen cells were thawed four days before imaging and plated only 24 h before imaging. These cells were a gift from the Laboratory for Fluorescence Dynamics, UC-Irvine. Fixed labeled BPAE cells were purchased from ThermoFisher (F36924); this slide contains bovine pulmonary artery endothelial cells (BPAEC). The mitochondria were labeled with MitoTracker Red CMXRos (before fixation), F-actin with Alexa FluorTM 488 phalloidin, and the nuclei with the blue-fluorescent DNA marker DAPI. The laser power was maintained below 25 mW for live-cell imaging. To generate additional contrast in the live cell experiments, we used a higher laser power that could induce apoptosis as shown in Fig. 6b. This power was set at 53.5 mW, and the laser power was controlled using an electro-optic modulator.

### Reporting summary

Further information on research design is available in the Nature Portfolio Reporting Summary linked to this article.

## Data availability

The minimum LiDAR and FLIM (simulated and real) datasets to run the code are available at https://github.com/JonghoLee0/CASPI. The Middlebury dataset and the CARLA simulator for the LiDAR application are available at https://vision.middlebury.edu/stereo/data/and https://carla.org/, respectively.

## Code availability

The code to reproduce the results is available at https://github.com/JonghoLee0/CASPI.

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

## Acknowledgements
This research was partially funded by an NSF CAREER Award 1943149 (M.G.), Wisconsin Alumni Research Foundation (M.G.), U.S. Department of Energy grant DOE-SC0019013 (K.W.E.), NIH U54CA268069 (K.W.E.), RRF Walter H. Helmerich Research Chair (KWE) and an NSF ERI Award 2138471 (A.I.).

## Author contributions
J.L. conceived and implemented the CASPI algorithm, performed the LiDAR and FLIM simulations, and applied the CASPI algorithm to all measurements in the paper and Supplementary Information. A.I. and M.G. took part in designing the LiDAR experiments. A.I. performed the LiDAR experimental measurements. J.V.C. and K.W.E. took part in designing the FLIM experiments. J.V.C. performed the FLIM experimental measurements. K.W.E. supervised the FLIM experiments. M.G. supervised all aspects of the project. All authors took part in writing the paper and Supplementary Information.

## Competing interests
M.G. is a co-founder and shareholder of Ubicept, a start-up company that is engaged in developing single-photon camera systems for various computer vision and imaging applications. K.W.E. is a co-founder of OnLume Inc., a start-up company that makes fluorescence guided surgery systems. K.W.E. is also a consultant to Bruker Inc. on their multiphoton microscopy development, and to Elephas Inc. on their platform for doing drug studies on patient biopsies. The remaining authors declare no competing interests.
