## [Peer Review File · Nature Communications]

CASPI: Collaborative Photon Processing for Active Single-Photon ImagingREVIEWER COMMENTS

Reviewer #1 (Remarks to the Author):

The paper describes a technique for processing single photon image data streams containing depth or fluorescence lifetime timing information to produce LIDAR or FLIM images with improved tolerance of background level or low photon counts. The paper is written and illustrated to a high standard.

I have three concerns relating to the work:

1. Some better comparative referencing. There is no mention of quanta image processing which appears to provide similar methods (see for example the recent review 10.1109/TED.2022.3166716 and the work of Chan et al. For lifetime image processing I see no mention or comparison with global analysis methods e.g. <https://doi.org/10.1364/OE.17.006493>. There is also a substantial literature on pile-up and in LIDAR and FLIM yet three references [22-24] are used (two self references).
2. The signal to background ratio (SBR) makes an assumption of a first photon time to digital converter with a long dead time. The SBR takes the sum of all photons from background and ratios that with the sum of photons in the detected laser pulse. However that does not give a useful notion of the limit of detectability. Moreover this SBR depends substantially on the precise pixel distance and reflectivity. Please state which pixel condition of the scene is used and why this is representative of the scene. A more useful definition of SBR is given in 10.1109/TIM.2021.3073684 and 10.1109/TED.2021.3131430. Please consider using this definition.
3. Eventually some of the limitations of first photon TDCs will be addressed by advanced CMOS integration involving faster sampling mechanisms such as multi-event TDCs and multi-bit gated pixels. The perspective of how CASPI could be influence on-chip processing is missing from the paper. Please provide some discussion.

Minor points:

1. How would the method deal with multipath returns in LIDAR?
2. Many of the example images e.g. Fig. S2 seem to lose high frequency information horizontally e.g. window frame and in that respect I prefer the grainy photon count image. Is there some underlying reason?

Reviewer #2 (Remarks to the Author):

This manuscript presents a novel method to extend the capabilities of single photon time-resolved cameras to low and high photon regimes. It uses a combination of techniques, such as gathering information across similar pixels, photon pile-up correction and frequency filtering.

It is very well written, has clear examples of the capabilities of the new CASPI algorithm, and is compared to existing state of the art techniques using qualitative and quantitative methods. Methods are well described and code is provided. The work is placed in context very well with good referencing of prior works. The work is timely and potentially of high impact to the field.

The Methods section mentions the use of an additional intensity image that can be used to improve the results. Please make clear in the manuscript if an intensity image was used in any of the examples.

It occurs to me that there could be potential limitations and effects that are not mentioned. Could the authors please comment on the following and if necessary add to the manuscript.

- 1) It seems that the local processing may act in some way like local binning of the data and may

cause blurring of image features in the spatial domain. Does this processing reduce the effective resolution of the image?

2) The process of combining data, both locally and non-locally, by finding "similar cubelets" may result in a form of pixel or cubelet clustering that may cause quantisation of the image into homogeneous regions. Does CASPI reduce the natural variations within the scene by clustering pixels with a *_similar_* response and forcing them to have the *_same_* response?

3) If the effects in 1 and 2 are present, it seems that there may be thresholds and size parameters that could change the performance. If so, could this be described in the manuscript, and can the method then be described as "tuning free"?

4) The Methods for FLIM describe a single lifetime for each pixel (Methods Equation 3), inferring that a mono-exponential response is assumed. Is CASPI tuned to mono-exponential responses, or will it retain multi-exponential responses?

Response to Reviewers' Comments on "CASPI: Collaborative Photon Processing for Active Single-Photon Imaging"

Jongho Lee, Atul Ingle, Jenu V. Chacko, Kevin W. Eliceiri & Mohit Gupta

We thank the reviewers for their feedback on our manuscript. This new version of the manuscript includes substantial revisions to the text, and includes additional results and analyses, as well as new figures, to address the weaknesses in our initial submission. We have highlighted relevant sections in the manuscript text with two different colors (**Reviewer 1** and **Reviewer 2**) to make it easier to locate all changes. We believe that the text and additional results in this revision will address Reviewer 1's comments about comparison to other approaches (quanta sensors, pile-up correction methods, global analysis) and Reviewer 2's comments about potential limitations of CASPI. A point-by-point response to both reviewers' comments are below.

Response to **Reviewer 1's** Comments

The paper describes a technique for processing single photon image data streams containing depth or fluorescence lifetime timing information to produce LIDAR or FLIM images with improved tolerance of background level or low photon counts. The paper is written and illustrated to a high standard. I have three concerns relating to the work

We thank the reviewer for their positive feedback about the writing and illustrations. We hope that the responses below and the substantial changes in this revised manuscript address the three major concerns and the two minor concerns.

Some better comparative referencing. There is no mention of quanta image processing which appears to provide similar methods (see for example the recent review 10.1109/TED.2022.3166716 and the work of Chan et al.

We agree with the reviewer that it is important to include comparative discussion and citations for quanta image processing approaches. We have added these in the "Introduction" section of the main text. Although CASPI and quanta image processing approaches share a common goal of recovering scene information in challenging illumination conditions, their target domains (and hence, the technical methods) are quite different: Quanta image sensors (QIS) and QIS data processing approaches recover high-quality 2D intensity images for *passive imaging* in low signal and high-dynamic range conditions. In contrast, CASPI is targeted toward robust recovery of 3D photon transient cubes in *active imaging* (e.g., 3D depth sensing, fluorescence lifetime imaging) under extreme lighting conditions. We have added citations for the review paper (Ma *et al.*) and the two papers by Chan *et al.* on page 4 of this revised manuscript and clarified the similarities and differences between CASPI

and QIS processing.

For lifetime image processing I see no mention or comparison with global analysis methods e.g. <https://doi.org/10.1364/OE.17.006493>.

We thank the reviewer for noting this missing comparisons with the global analysis method of Grecco *et al.* which is a calibration-free method for single- and multi-exponential FLIM decays in low SNR conditions. We have now added a new section titled “Comparisons with Global Analysis for Multi-exponential Decay in FLIM” in the revised manuscript (page 14) and new results in Fig. 7. We estimate the relative contributions of the two exponentials in a bi-exponential decay model. We assume that the two individual lifetimes are fixed across the FLIM image, but their relative contributions vary spatially. We compare three methods: (a) pixel-wise fitting with raw measurement, (b) global fitting with raw measurements, and (c) CASPI. As shown in Fig. 7 of the revised manuscript, pixel-wise fitting with CASPI gives the best accuracy among the three methods even with as few as 200 signal photons per pixel.

There is also a substantial literature on pile-up in LIDAR and FLIM yet three references [22-24] are used (two self references).

We agree with the reviewer that the problem of pileup mitigation in active single-photon imaging has received considerable interest recently. To provide a reader with a more comprehensive summary of research in this area, we have now included additional citations on computational and hardware approaches for pileup mitigation in both LiDAR and FLIM. These can be found in the “Introduction” section of the revised manuscript (refs. 25–28).

The signal to background ratio (SBR) makes an assumption of a first photon time to digital converter with a long dead time. The SBR takes the sum of all photons from background and ratios that with the sum of photons in the detected laser pulse. However that does not give a useful notion of the limit of detectability. Moreover this SBR depends substantially on the precise pixel distance and reflectivity. Please state which pixel condition of the scene is used and why this is representative of the scene. A more useful definition of SBR is given in [10.1109/TIM.2021.3073684](https://doi.org/10.1109/TIM.2021.3073684) and [10.1109/TED.2021.3131430](https://doi.org/10.1109/TED.2021.3131430). Please consider using this definition.

We agree with the reviewer that it is important to use an SBR measure that captures a useful notion of the limit of detectability. The incident-photon-ratio definition has been used in recent literature on single-photon LiDAR¹. In the revised manuscript we have added text on page 8 to clarify why we decided to use the incident-photon-ratio definition of SBR, and its relationship with the limits of detectability.

Useful notion of detectability: We believe the incident photons definition of SBR captures a useful notion of detectability for two reasons. (a) Incident photon counts are a better indicator of the true scene illumination conditions as compared to the detected photon counts. This is due to the fact that single-photon cameras detect only the first incoming photon (while rejecting subsequent photons in each illumination cycle). (b) In the low photon flux regime, the incident photon SBR is equivalent to the detected photons SBR definition¹.

Relationship to the SBR definition given in 10.1109/TIM.2021.3073684: The detected-photon-ratio definition of SBR used in previous LiDAR literature^{2,3,4} assumes a low photon flux regime where effects of dead-time and pileup can be neglected. In such scenarios, the detected-photon-ratio SBR measure is practically the same as the incident-photon-ratio SBR measure¹ used in our manuscript. However, in high photon flux regimes (e.g., strong ambient light), the detected-photon-ratio definition of SBR used in previous literature may not capture the scene illumination conditions due to the pileup effects as described above. In these settings, the incident-photon-ratio definition provides a more accurate notion of detectability. We have added this clarification on page 8 of this revised manuscript.

Relationship to the SBR definition for a Gaussian pulse model in 10.1109/TED.2021.3131430: Since CASPI can be generalized to various applications that deal with non-Gaussian pulse shapes as well (e.g., LiDAR systems with asymmetrical laser pulse shapes, FLIM with mono- or multi-exponential IRFs, or transient imaging with multiple peaks), we start with a more general definition of SBR that can handle arbitrary pulse shapes. The definition of SBR given in 10.1109/ TED.2021.3131430 uses a Gaussian laser pulse model and expresses SBR in terms of the signal and background photons and the standard-deviation (or the full-width at half-maximum) of the Gaussian IRF. We have accounted for this Gaussian pulse model in Eq. (18) in the Methods section of this revised manuscript.

Eventually some of the limitations of first photon TDCs will be addressed by advanced CMOS integration involving faster sampling mechanisms such as multi-event TDCs and multi-bit gated pixels. The perspective of how CASPI could be influence on-chip processing is missing from the paper. Please provide some discussion.

We agree with the reviewer that photon throughput and pileup limitations with today's TDCs could be mitigated to a certain extent in the future through hardware improvements. We believe that CASPI is complementary to hardware innovation and can be used to resolve the limitations of not only the first photon detection in photon-flooded regime, but also low photon counts in photon-starved regime as shown in Figs. 3, 4, 5, 6, and 7 — when there are no/minimal pileup distortions. Low SNRs caused by low photon counts is a fundamental problem that often occurs due to long scene distances, low scene albedo, low source power, etc. We have added a discussion about this in “Limitations and Discussion” of this revised manuscript (pages 15 and 16) and also in the “Methods” section (page 18).

Minor point 1: How would the method deal with multipath returns in LIDAR?

Since CASPI does not impose any priors on the number of reflections or shape of the underlying photon transients, our method can be used to recover multi-path transients as well. This may be important for applications such as transient imaging where the entire waveform needs to be recovered, or in resolving multiple bounce returns in a LiDAR. Further, although multipath reflections can cause large systematic depth errors for conventional LiDAR imaging, those indirect illumination components can be exploited to recover 3D shapes out of the direct line-of-sight⁵. In this revision, we have added a new section titled “Recovering Multipath Transients” (see page 11). We show that CASPI can recover indirect illumination components in addition to direct components even with low signal photon counts. Figure 5 in the revised manuscript shows the simulation results with an indoor scene where CASPI is able to recover the complete temporal profiles of light transport with as few as 10 signal photons/pixel on average.

Minor point 2. Many of the example images e.g. Fig. S2 seem to lose high frequency information horizontally e.g. window frame and in that respect I prefer the grainy photon count image. Is there some underlying reason?

CASPI does not have any preferential orientation for recovering high-frequency information. As shown in Response Fig. 1 below, the horizontal lines were recovered, albeit with lower contrast. We believe the high-frequency information loss in this case happened because it is challenging to distinguish between small signal structures and noise in this extreme sub-photon regime (0.2 signal photons/pixel), and the non-local aggregation emphasized the vertical edges that are more dominant than the horizontal edges in this window frame.

Response Figure 1: Magnified view of Laundry scene in Fig. S2. The horizontal edges in the window frame are visible in the zoomed insets. The vertical edges in the window frame are emphasized in the CASPI reconstruction likely due to the fact that the vertical edges are more dominant than the horizontal edges in the raw photon data.

Response to Reviewer 2's Comments

This manuscript presents a novel method to extend the capabilities of single photon time-resolved cameras to low and high photon regimes. It uses a combination of techniques, such as gathering information across similar pixels, photon pile-up correction and frequency filtering. It is very well written, has clear examples of the capabilities of the new CASPI algorithm, and is compared to existing state of the art techniques using qualitative and quantitative methods. Methods are well described and code is provided. The work is placed in context very well with good referencing of prior works. The work is timely and potentially of high impact to the field.

We thank the reviewer for their positive feedback about the potential impact of this work.

The Methods section mentions the use of an additional intensity image that can be used to improve the results. Please make clear in the manuscript if an intensity image was used in any of the examples.

The only result in the paper that uses an additional intensity image is shown in Fig. S3 in Supplementary Results which demonstrates how depth estimation results can be improved when intensity images are available as additional input. None of the other results (including those shown in the main text) rely on the additional intensity image input. We appreciate the reviewer raising this question because it was not clearly mentioned in the first version of our manuscript. We have now added a clarification in the “Scene Intensity Images with CASPI” section of the revised manuscript on pages 7 and 8.

It occurs to me that there could be potential limitations and effects that are not mentioned. Could the authors please comment on the following and if necessary add to the manuscript.

1) It seems that the local processing may act in some way like local binning of the data and may cause blurring of image features in the spatial domain. Does this processing reduce the effective resolution of the image?

In the revised manuscript we have included additional discussion to emphasize the difference between CASPI and spatial binning methods. In particular, we have added new text to the section titled “Hierarchical Blind Photon Processing” (pages 6 and 7) emphasizing the difference between CASPI and spatial binning. The frequency-domain processing used in CASPI can reliably separate signal and noise components because any structured low-frequency signal components accumulate better than random noise components. We show this in Fig. 2 in the main text. If the amount of noise relative to the signal is sufficiently reduced by local correlations, CASPI can also recover *high-frequency* signal components using the proposed noise estimation step. In contrast, spatial binning only uses local spatial averaging which is limited to the low-frequency (DC) com-

ponents in the frequency domain. Hence, CASPI can maintain high-frequency spatial details better than naïve spatial binning.

This difference in performance is demonstrated via experimental comparisons between CASPI and spatial binning, as shown in Figs. 6, S5, S6, and S7 in this revised version of the manuscript. In the presence of extremely strong noise, we do expect some loss of spatial information and reduction in the effective resolution of the recovered photon transient cube due to the local processing of CASPI (recovering photon transients using only local correlations). However, these results show that CASPI preserves the effective resolution better than conventional methods that rely on spatial binning.

*2) The process of combining data, both locally and non-locally, by finding "similar cubelets" may result in a form of pixel or cubelet clustering that may cause quantisation of the image into homogeneous regions. Does CASPI reduce the natural variations within the scene by clustering pixels with a *_similar_* response and forcing them to have the *_same_* response?*

If similar cubelets are simply averaged, it may result in quantized homogeneous regions as the reviewer pointed out because the recovered photon transients will be blurred and any subtle structural difference between these similar cubelets will be lost. However, CASPI is quite different from simple averaging. CASPI effectively separates the signal and noise components in the frequency domain through *accurate noise estimation*. It prevents blurring and retains high-frequency components corresponding to the subtle structural difference between the similar cubelets while also suppressing the noise components effectively. We clarified this point in the ‘Hierarchical Blind Photon Processing’ Section (page 7) of the revised manuscript.

3) If the effects in 1 and 2 are present, it seems that there may be thresholds and size parameters that could change the performance. If so, could this be described in the manuscript, and can the method then be described as "tuning free"?

We had used the term “tuning-free” to indicate that CASPI performs accurate noise estimation without requiring any prior knowledge of noise statistics. However, we agree with the reviewer that this term may cause a misunderstanding that CASPI does not have any tuning parameters. CASPI has several parameters such as the cubelet size, the search window size, and the SNR threshold for adaptive photon processing in initial flux estimation. Quite remarkably, we found that these parameters do not need much manual tuning across datasets. The parameter values mentioned in the paper work for a wide range of operating illumination conditions, for both simulated and experimental data for both FLIM and LiDAR. We used the same parameter values for all results shown in the paper, except for Fig. 6a, where we used a smaller cubelet size for better recovery of the small mitochondrial structures. In the revised manuscript, we have removed the term “tuning-free” and instead

refer to our method as being “blind” to contrast it from non-blind denoising methods that require the user to provide the noise level.

4) The Methods for FLIM describe a single lifetime for each pixel (Methods Equation 3), inferring that a mono-exponential response is assumed. Is CASPI tuned to mono-exponential responses, or will it retain multi-exponential responses?

We thank the reviewer for mentioning the important issue of estimating multi-exponential decay profiles for FLIM. CASPI is a model-free and prior-free algorithm, and hence not restricted to mono-exponential decay models. We have added new results and discussion in the section titled “Comparisons with Global Analysis for Multi-exponential Decay in FLIM” in this revised manuscript (page 14). We estimated the relative contributions of the bi-exponential decay model (two lifetimes are fixed over the FLIM image) by pixel-wise fitting with raw measurement, global fitting with raw measurement, and pixel-wise fitting with CASPI. As shown in Fig. 7, pixel-wise fitting with CASPI can recover bi-exponential decay parameters with as few as 200 signal photons per pixel. These results show that CASPI is not limited to mono-exponential decay models.

References

1. Rapp, J., Ma, Y., Dawson, R. M. & Goyal, V. K. High-flux single-photon lidar. *Optica* **8**, 30–39 (2021).
2. Koerner, L. J. Models of direct time-of-flight sensor precision that enable optimal design and dynamic configuration. *IEEE Transactions on Instrumentation and Measurement* **70**, 1–9 (2021).
3. Rapp, J. & Goyal, V. K. A few photons among many: Unmixing signal and noise for photon-efficient active imaging. *IEEE Transactions on Computational Imaging* **3**, 445–459 (2017).
4. Lindell, D. B., O’Toole, M. & Wetzstein, G. Single-photon 3d imaging with deep sensor fusion. *ACM Trans. Graph.* **37**, 113–1 (2018).
5. Velten, A. *et al.* Recovering three-dimensional shape around a corner using ultrafast time-of-flight imaging. *Nature communications* **3**, 1–8 (2012).

REVIEWER COMMENTS

Reviewer #1 (Remarks to the Author):

I thank the authors for their responses to my comments. I am happy with most of the revisions apart from the part relating to SBR. I looked up the authors own definition of SBR (cut and pasted from their Optica reference).

fluorescence lifetime in FLIM or additional reflections for multidepth imaging. We refer to the quantities $S := \int_0^t \lambda_s(t) dt$, $B := \lambda_b t_r$, and $\Lambda := \int_0^t \lambda(t) dt = S + B$ as the signal flux, background flux, and total flux, respectively, which represent the mean number of photoelectrons generated by each process within one illumination cycle. The flux could be considered equivalent to the detection rate if there were no dead times. The signal-to-background ratio (SBR) is S/B .

The issue I see with this definition is that it must be parameterised with a suitable integration time t . This is presumably a multiple of a cycle time of the laser or at least a fraction of that time in which photons are being gathered by the system. Thus if the laser repetition rate is reduced resulting in a long period between pulses, more background photon counts will be gathered as B . The signal counts will be unaltered if the laser peak power is unchanged. Thus with this definition the SBR reduces as the repetition rate of the laser is modified. However, the actual detectability limit of the system is only related to the signal photons received from the reflected laser pulse and the number of background photons received during the time duration of the laser pulse. This is the SBR definition related to the work from Koerner and Gyongy.

It is for that reason that quoting a SBR (as found in the paper in review) of say 3000/3000 in high background conditions still does not give a good impression of how far we are from a detectability limit. By the authors definition, this is a SBR=1. However, take an example of a LIDAR system with a laser pulse duration of 100ps and with a 100ps TDC bin and say 3000 bins (for simplicity), then that is 1 background *incident* photon on average in every histogram bin but 3000 *incident* photons in the signal peak bin. Looking at the SBR in the laser pulse duration that is actually an SBR=3000/1. That is very far above detectability. The detectability limit is closer to a few photons (6~7 for 99.9% detectability) in the peak signal return bin.

There is no mention here about a system-related issues such as pile-up, SPAD dead time. SPAD detection efficiency and TDC dead time. After these processes, the SBR will indeed be further degraded. However, I see no reason the *incident* SBR need be parameterised by a system

dependent integration time. I'm not even sure of the value of that integration time in producing the numbers in the SBR ratios in the paper. As a result, I'm not happy with the authors pointing to their own definition of SBR in justification without discussing these issues and hinging the whole paper on some SBR numeric ratios.

Reviewer #2 (Remarks to the Author):

I thank the authors for the updated manuscript. All of my concerns and questions have been addressed.

Response to Reviewers' Comments on "CASPI: Collaborative Photon Processing for Active Single-Photon Imaging"

Jongho Lee, Atul Ingle, Jenu V. Chacko, Kevin W. Eliceiri & Mohit Gupta

Response to **Reviewer 1**'s Comments on SBR Definition

I thank the authors for their responses to my comments. I am happy with most of the revisions apart from the part relating to SBR. I looked up the authors own definition of SBR (cut and pasted from their Optica reference). ... The issue I see with this definition is that it must be parameterised with a suitable integration time t . This is presumably a multiple of a cycle time of the laser or at least a fraction of that time in which photons are being gathered by the system. Thus if the laser repetition rate is reduced resulting in a long period between pulses, more background photon counts will be gathered as B . The signal counts will be unaltered if the laser peak power is unchanged. Thus with this definition the SBR reduces as the repetition rate of the laser is modified. However, the actual detectability limit of the system is only related to the signal photons received from the reflected laser pulse and the number of background photons received during the time duration of the laser pulse. This is the SBR definition related to the work from Koerner and Gyongy.

We agree with the reviewer that the signal-to-background ratio defined using the photon counts during the signal pulse width is an important measure of the detectability limit of the sensor. In this revised manuscript, we now show the pulse signal-to-noise ratio metric:

$$\text{SBR}_{\text{pulse}} = \frac{N_{\text{sig}}}{n_{\text{bkg}}}$$

where N_{sig} and n_{bkg} denote the number of signal photons and the number of background photons received during the laser pulse peak duration. This is the SBR definition given in Eq. (6) of Gyongy et al.¹ As pointed out by the reviewer, $\text{SBR}_{\text{pulse}}$ is independent of the integration time used during acquisition, and for a fixed pulse energy, does not vary with the laser repetition period.

It is for that reason that quoting a SBR (as found in the paper in review) of say 3000/3000 in high background conditions still does not give a good impression of how far we are from a detectability limit. By the authors definition, this is a SBR=1. However, take an example of a LIDAR system with a laser pulse duration of 100ps and with a 100ps TDC bin and say 3000 bins (for simplicity), then that is 1 background incident photon on average in every histogram bin but 3000 incident photons in the signal peak bin. Looking at the SBR in the laser pulse duration that is actually an SBR=3000/1. That is very far above detectability. The detectability limit is closer to a few photons (6-7 for 99.9% detectability) in the peak signal return bin.

For active single-photon imaging, modeling the background photon counts *outside the signal pulse duration* is important; it is well known that under strong ambient illumination conditions (e.g., bright sunlight for an outdoor LiDAR), such background photons outside the pulse cause strong pileup distortions, resulting in large and systematic errors in recovered depths.

In particular, the background photons outside the signal pulse width increase the chance of false signal peak detection and lower the performance of computational LiDAR imaging approaches due to pileup from ambient light. This effect is especially pronounced in outdoor long-range imaging where the laser peak arrives close to the end of a laser period. An example of this effect can be seen in the “Outdoor” LiDAR result shown in Fig. 3a (last row) and an example photon transient in Fig. 3b (last column). Note that even with a seemingly high $SBR_{\text{pulse}} = 200$, existing computational methods fail to reconstruct farther scene points such as the building in the distance due to pileup from early arriving ambient photons. In such conditions, it is necessary to use an SBR definition that includes not just the background photons within the laser pulse, but the total number of background photons over the *entire laser cycle duration*. Therefore, several computational LiDAR imaging approaches^{2,3,4} report SBR as a ratio of two numbers: the signal and background photon counts over a full laser period instead of just the signal pulse width. In our revised manuscript report this using the total SBR:

$$SBR_{\text{total}} = \frac{N_{\text{sig}}}{N_{\text{bkg}}}$$

where N_{sig} and N_{bkg} denote the total number of signal and background photons received over the entire exposure time. Note that although SBR_{total} is a ratio of two numbers, the absolute values of N_{sig} and N_{bkg} are important to report. For example, an $SBR_{\text{total}} = 1/1$ is not the same operating condition as $SBR_{\text{total}} = 3000/3000$ even though both ratios reduce to 1.0. The latter is a high background flux regime susceptible to strong pileup due to ambient light, and therefore, considerably more challenging for computational LiDAR methods.

Prior work on FLIM also uses the total signal photon counts (over the total number of laser cycles) to describe the conditions for reliable fluorescence lifetime estimation. According to some estimates^{5,6}, 100 photons/pixel and 1,000 photons/pixel are required for mono-exponential and bi-exponential decay models. Reporting SBR_{total} enables a fair comparison between results obtained with CASPI and those found in previous FLIM papers.

Please refer to the orange text on pages 8 and 10, Tables S1 and S2, and Figures 3, 4, S2, and S3 in the revised manuscript for explanations on the two definitions of the SBR, and the corresponding values for various illumination conditions for LiDAR results.

There is no mention here about a system-related issues such as pile-up, SPAD dead time. SPAD detection efficiency and TDC dead time. After these processes, the SBR will indeed be further degraded. However, I see no reason the incident SBR need be parameterised by a system dependent integration time. I'm not even sure of

the value of that integration time in producing the numbers in the SBR ratios in the paper. As a result, I'm not happy with the authors pointing to their own definition of SBR in justification without discussing these issues and hinging the whole paper on some SBR numeric ratios.

We now include a discussion about the issue of ambient light pileup and the importance of using SBR_{total} metric on page 8 of the revised manuscript. We also provide the values of the different integration times in Tables S1-S4 that were used to compute the $N_{\text{sig}}/N_{\text{bkg}}$ numeric ratios for SBR_{total} .

References

1. Gyongy, I., Dutton, N. A. & Henderson, R. K. Direct time-of-flight single-photon imaging. *IEEE Transactions on Electron Devices* **69**, 2794–2805 (2021).
2. Rapp, J. & Goyal, V. K. A few photons among many: Unmixing signal and noise for photon-efficient active imaging. *IEEE Transactions on Computational Imaging* **3**, 445–459 (2017).
3. Rapp, J., Ma, Y., Dawson, R. M. & Goyal, V. K. High-flux single-photon lidar. *Optica* **8**, 30–39 (2021).
4. Lindell, D. B., O'Toole, M. & Wetzstein, G. Single-photon 3d imaging with deep sensor fusion. *ACM Trans. Graph.* **37**, 113–1 (2018).
5. Yang, H. *et al.* Protein conformational dynamics probed by single-molecule electron transfer. *Science* **302**, 262–266 (2003).
6. Gratton, E., Breusegem, S., Sutin, J. D., Ruan, Q. & Barry, N. P. Fluorescence lifetime imaging for the two-photon microscope: time-domain and frequency-domain methods. *Journal of biomedical optics* **8**, 381–390 (2003).

REVIEWERS' COMMENTS

Reviewer #1 (Remarks to the Author):

The clearer definitions and discussion of SBR address my comments. No further remarks.